# SPRY domains encode ubiquitin ligase specificity for ZAP and RIG-I

Ibrahim Syed[1], Sheng Chen[1], David J. Peeler[1,2], Paul F. McKay[1], Marco A. Briones-Orta[1], Jennifer A. Bohn[3], Robin J. Shattock[1], Daniel Gonçalves-Carneiro[1]*

1 Department of Infectious Diseases, Faculty of Medicine, Imperial College London, London, United Kingdom, 2 Department of Physiology, Anatomy and Genetics, Department of Engineering Science, Kavli Institute for Nanoscience Discovery, University of Oxford, Oxford, United Kingdom, 3 RockEDU Science Outreach, The Rockefeller University, New York, New York, United States of America

* d.goncalves-carneiro@imperial.ac.uk

## Abstract

Innate immune sensors rely on ubiquitin ligases to calibrate antiviral responses, yet the rules governing substrate recognition by SPRY-containing ligases remain poorly defined. Here, we establish a large-scale structure-based screening pipeline using AlphaFold to systematically predict interactions between human nucleic acid sensors and SPRY-containing proteins. Our approach uncovered novel transient or degradation-sensitive interactions that are typically missed by proteomic methods, including a labile TRIM58-OAS1 complex. We show that SPRY domains dictate substrate specificity: TRIM25 preferentially engages ZAP, whereas Riplet favors RIG-I. Domain-swapping experiments demonstrated that SPRY domains are sufficient to reprogram ligase specificity and antiviral activity. Phylogenetic and structural analyses revealed that TRIM25 and Riplet evolved from a common ancestor but diverged in coiled-coil architecture and oligomeric state, while retaining conserved substrate preferences. Residue-level modeling identified hypervariable SPRY loops as critical determinants of recognition, a prediction validated by targeted mutagenesis of the TRIM25-ZAP interface. Finally, we show that distinct SPRY-containing ligases surveil self-amplifying RNA (saRNA) vaccines: Riplet-RIG-I primarily responds when RNA is delivered by lipofection, whereas TRIM25-ZAP is engaged upon lipid nanoparticle delivery, with functional consequences for vaccine expression. Together, these findings demonstrate that SPRY domains encode recognition logic for ubiquitin ligases, that AlphaFold enables discovery of otherwise hidden interactions and that these principles have direct implications for RNA-based therapeutics.

## Author summary

The innate immune system must respond rapidly to viral infection while avoiding excessive activation that can damage host tissues and promote inflammatory disease. A major way cells maintain this balance is through ubiquitin ligases,

**Data availability statement:** All data generated in this study will be provided as supplementary files. Scripts used to calculate interaction scores and to plot PAE maps can be found on github.com/DanSallves/ProteinInteraction/.

**Funding:** This study was funded by the Medical Research Council (MRC, MR/Z50421X/1 awarded to D.G.-C.), the Imperial College Research Fellowship (awarded to D.G-C), the Engineering and Physical Sciences Research Council (EPSRC, EP/Y010167/1 and EP/Y530529/1 awarded to R.J.S.) and the NIHR Imperial Biomedical Research Centre (BRC, awarded to R.J.S.). D.G-C received salary from Imperial College Research Fellowship and the MRC. The findings presented in this article are made by the authors. The funders had no role in study design, data collection and analysis, decision to publish, or preparation of the manuscript.

**Competing interests:** The authors have declared that no competing interests exist.

enzymes that tune the activity, stability and localization of antiviral proteins. However, it has remained unclear how these ligases selectively recognize the correct immune targets. Here, we used a large-scale AlphaFold-based structural screening strategy to identify interactions between human nucleic acid sensors and SPRY-containing proteins, a relatively poorly understood group of immune regulators. This approach revealed interactions that are difficult to detect experimentally because they are transient or unstable. We found that SPRY domains determine which antiviral sensors are recognized by different ligases, and that swapping these domains is sufficient to redirect immune specificity and function. We also identified key structural features that control this recognition. These findings are important because they explain a fundamental mechanism by which innate immunity is both activated and restrained. They also show how structure-based prediction can uncover hidden regulatory interactions with direct relevance for RNA-based therapeutics, including self-amplifying RNA vaccines, whose activity is shaped by the way innate immune sensors detect them.

## Introduction

Innate immunity relies on a finely tuned balance between responsiveness and restraint. Insufficient activation permits unchecked viral replication, while excessive or misdirected responses can result in inflammation and autoimmunity. A major mechanism by which cells calibrate this balance is through ubiquitin (Ub)-mediated regulation of immune effectors [1]. Ubiquitination modulates protein activity, localization and stability, enabling dynamic control over signaling cascades. At the center of this system are E3 ligases, which confer substrate specificity within the Ub conjugation cascade [2]. Among them, TRIM (tripartite motif) have emerged as central players in antiviral defense, many of which utilize C-terminal PRY/SPRY (SPRY) domains to engage substrates [3]. Despite the presence of nearly 100 SPRY-containing proteins in the human genome, the rules that govern substrate recognition remain poorly defined.

This regulatory mechanism is exemplified by many RNA sensors, as diverse as retinoic acid-inducible gene I (RIG-I) and the zinc finger antiviral protein (ZAP). RIG-I detects 5′-triphosphorylated RNA and signals through MAVS to induce type I interferon production [4,5], while ZAP binds CpG-rich viral RNAs and recruits cofactors to degrade or translationally repress them [6–8]. Both sensors are regulated by multiple E3 ligases: TRIM25 and Riplet catalyze ubiquitination of RIG-I and ZAP to enhance antiviral activity [9–13], whereas other ligases can target sensors for proteasomal degradation [14,15]. Thus, the same class of enzymes that promote antiviral immunity can also constrain it, either by modulating activation thresholds or by removing sensors to prevent chronic stimulation. However, these interactions are often transient or unstable, rendering them difficult to capture with high-throughput methods, such as affinity purification coupled with mass spectrometry.

Substrate recognition by SPRY domains is further complicated by their structural variability. SPRY domains share a conserved β-sandwich core but differ substantially in their surface-exposed loops, which are regions that often dictate binding specificity [16,17]. These hypervariable loops have evolved to recognize a broad range of target proteins and minor sequence changes in these loops can dramatically alter substrate preference [18], yet the lack of a universal binding motif has hindered predictive modeling.

Recent advances in structure-based prediction, particularly with AlphaFold2 and AlphaFold-Multimer [19,20], offer a promising solution to these limitations. These tools can generate highly accurate models of individual proteins and protein complexes directly from sequence, including transient or degradation-sensitive interactions. Unlike sequence-based approaches, AlphaFold predictions incorporate three-dimensional complementarity at the interface, making it possible to detect interactions that elude experimental capture. For SPRY domains, where specificity is often encoded in flexible loops, structure-based prediction offers an unprecedented opportunity to decode recognition principles.

In this study, we developed a large-scale AlphaFold-based screening pipeline to systematically identify interactions between human nucleic acid sensors and SPRY-containing proteins. By integrating interface-level metrics, we prioritized putative complexes for experimental validation. Our screen recovered known interactions, validating the predictive framework, but also uncovered novel transient interactions. Co-immunoprecipitation assays confirmed several AlphaFold predictions, including complexes that are otherwise unstable or subject to rapid degradation. AlphaFold-based structural predictions also revealed that SPRY domains encode specific substrate preferences, with TRIM25 selectively recognizing ZAP and Riplet favoring RIG-I. Swapping the SPRY domains between these ligases was sufficient to reprogram their binding specificity and activity. Residue-level modeling identified conserved loop residues critical for TRIM25-ZAP interaction, which we validated through targeted mutagenesis. Extending these insights to a therapeutic context, we found that different SPRY-containing proteins can sense self-amplifying RNA (saRNA) vaccines in a delivery-dependent manner, impacting antigen expression and the subsequent innate immune response. Collectively, these findings demonstrate that SPRY domains act as determinants of immune recognition and that AlphaFold enables the identification of transient, functionally relevant interactions with direct implications for RNA-based therapeutics.

## Results

### Predicting interaction partners of SPRY domains

To systematically identify interactions between ubiquitin ligases and innate immune proteins involved in the early recognition of viral nucleic acids, we developed a large-scale in silico screen leveraging AlphaFold-based structural modeling. We constructed a computational pipeline to predict interactions between "bait" proteins, comprising all annotated human nucleic acid sensors (e.g., RIG-I, OAS1) and signal transducers implicated in antiviral defense (e.g., IRF3, TBK1), and a set of "prey" proteins consisting of all human proteins annotated to contain SPRY domains (Fig 1a). Most human SPRY domain-containing proteins are known to either have ubiquitin ligase activity or to act as substrate adaptors in ubiquitin-conjugation complexes (S1a Fig). Therefore, we hypothesize that many of these proteins participate in ubiquitination reaction during immune responses. For each predicted complex, we computed an interaction score derived from the cumulative predicted aligned error (PAE) at the protein-protein interface, normalized by protein length (see Methods). A summary of all computed interaction scores is shown in Fig 1b. As anticipated, most SPRY proteins exhibited low interaction scores (<1) across the panel of baits (S1b Fig); however, every SPRY protein tested yielded at least two high-confidence interactions (Fig 1b) with a small of group of SPRY-protein predicted to interact with multiple substrates (S1c Fig). To benchmark our predictive framework, we queried the dataset for previously characterized interactions and found that known SPRY-domain interactions with nucleic acid sensors exhibited interaction scores exceeding 2.5 (Fig 1c). Based on this empirical threshold, we adopted an interaction score of >2.5 as the cutoff for identifying candidate interactions for further validation. Among the top-scoring predictions, we observed a putative interaction between STING, which functions as a central adaptor in the cGAS-cGAMP pathway in DNA sensing, and the transmembrane protein

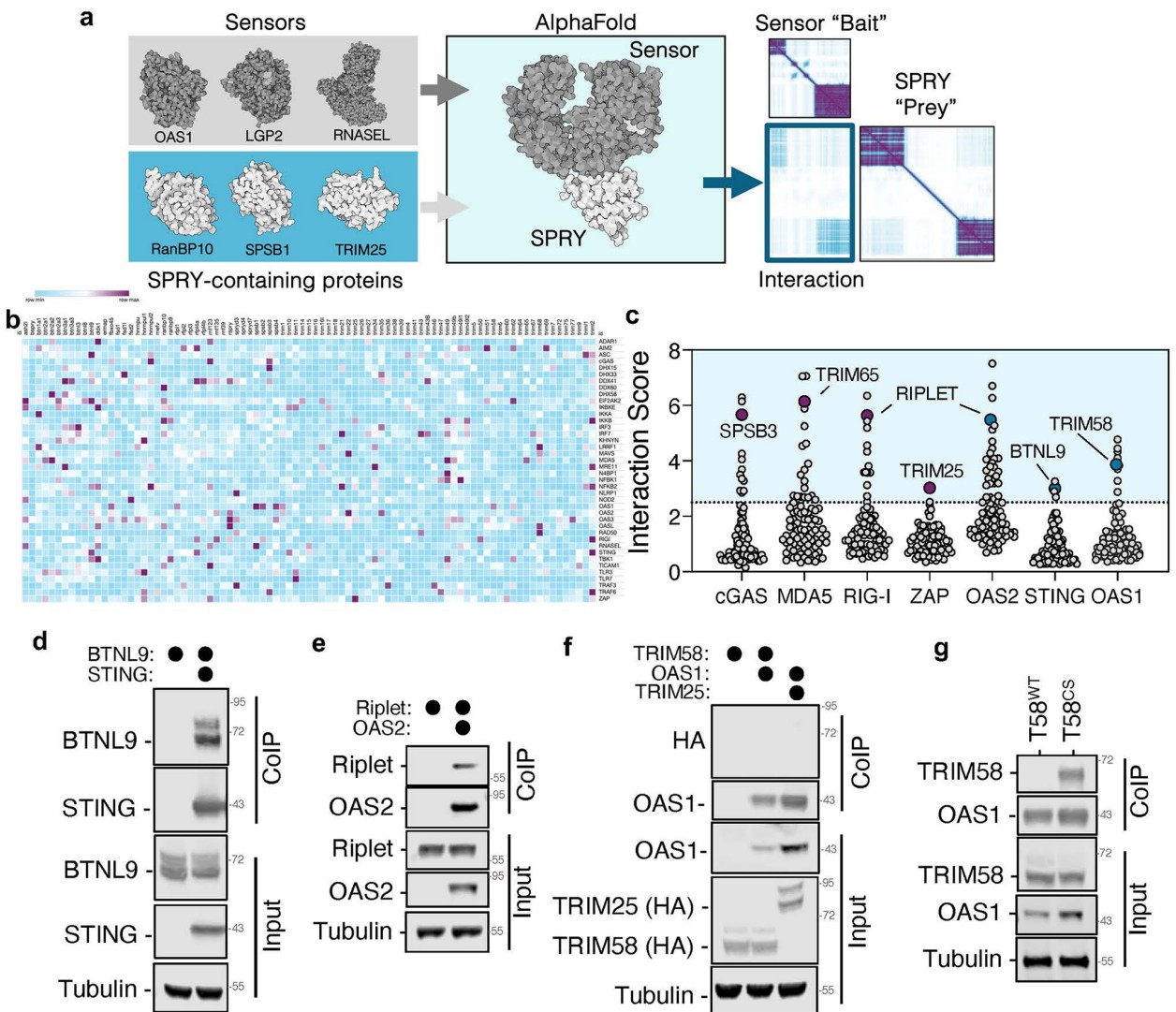

**Fig 1. SPRY domain-containing proteins interact with nucleic acid sensors.** Schematic representation of the pipeline to generate structural predictions of nucleic acid sensors and SPRY-containing proteins (a). Summary of all interaction scores observed between nucleic acid sensors and SPRY-containing proteins; interaction scores above the defined threshold are marked in purple (b). Distribution of interaction scores for cGAS, MDA5, RIG-I, ZAP, OAS2, STING and OAS1, with relevant known interactors (in purple) or novel validated interactors (in blue) indicated (c). HEK293T cells were co-transfected with plasmid encoding STING-FLAG and BTNL9-HA (d), Riplet-HA and OAS2-FLAG (e), or TRIM58-HA, TRIM25-HA and OAS1-FLAG (f). Many of these SPRY-containing proteins are detected as double bands on western blots, believed to be the result of non-degradative ubiquitination [21]. Protein complexes were immunoprecipitated from lysates with an anti-FLAG antibody and analyzed by western blotting. HEK293T cells were transfected with plasmids encoding wildtype or mutant TRIM58 (C57S,C60S) and OAS1-FLAG, followed by immunoprecipitation with anti-FLAG antibodies and western blot analysis (g). Western blots presented are representative of three independent experiments.

butyrophilin-like 9 (BTNL9) (Figs 1c and S1d Fig). Indeed, co-immunoprecipitation experiments validated a specific interaction between STING and BTNL9 in cells (Fig 1d). Similarly, one of the highest interaction scores observed for 2′-5′-oligoadenylate synthetase 2 (OAS2), a double-stranded RNA sensor, is Riplet (Figs 1c and S1c Fig); indeed, co-immunoprecipitation experiments revealed that OAS2 co-immunoprecipitated with Riplet (Fig 1e). We next examined a predicted interaction between 2′-5′-oligoadenylate synthetase 1 (OAS1) and the SPRY-containing E3 ubiquitin ligase

TRIM58, which also scored highly in our screen (Fig 1c). While we did not detect stable co-immunoprecipitation of OAS1 with TRIM58, we observed that OAS1 expression was substantially reduced in the presence of TRIM58, but not when co-expressed with a lowly scored SPRY-domain protein, TRIM25 (Figs 1f and S1e Fig). This observation raised the possibility that TRIM58 targets OAS1 for proteasomal degradation. To test this hypothesis, we generated a TRIM58 mutant lacking E3 ubiquitin ligase activity (TRIM58 C57S C60S, S1f Fig). Strikingly, this catalytically inactive TRIM58 variant robustly co-immunoprecipitated with OAS1 (Fig 1g), indicating that the TRIM58-OAS1 interaction occurs in cells but is likely transient and rapidly resolved through TRIM58-mediated degradation of OAS1. Together, these findings demonstrate that structure-based prediction using AlphaFold can identify both known and previously unrecognized protein-protein interactions, including labile or degradation-sensitive interactions that may escape detection in conventional assays.

### Sequence similarity does not predict SPRY-sensor interactions

We next asked whether SPRY domains with high sequence similarity are more likely to recognize the same substrates. First, we examined the contacting residues in highly scored interactions to define how SPRY-containing proteins engage their substrates. More than 65% of contact residues mapped to the SPRY domain (Fig 2a), indicating that this domain provides the major interaction interface, although other regions of the protein can also contribute. We therefore tested whether sequence similarity among SPRY domains correlated with the binding profile of each SPRY protein across the sensor panel. Overall, SPRY sequence identity correlated only weakly with similarity in predicted binding profiles (Fig 2b). Consistent with this, proteins with nearly identical SPRY domains did not necessarily display equivalent target profiles (Figs 2c and S2f Fig). These observations suggested that substrate recognition is not determined solely by overall SPRY-domain homology and that regions outside the SPRY domain may modulate binding specificity. To test this idea, we repeated the AlphaFold screen using only the SPRY domains of the prey proteins and compared the resulting interaction profiles with those obtained using full-length proteins. Across the two datasets, interaction scores from isolated SPRY domains and full-length proteins showed moderate overall concordance (S2a Fig), and many high-confidence interactions identified in this study were recovered in both analyses (S2b Fig). However, some interactions were detected only in the full-length dataset or only in the SPRY-only dataset, indicating that full-length context can influence predicted binding. For example, hnRNPUL2 contains an N-terminal SAP domain, a central SPRY domain and a C-terminal NTPase-like domain (S2c Fig). In the predicted hnRNPUL2:cGAS complex, the interaction interface includes residues from both the SPRY and NTPase-like domains, whereas the isolated SPRY domain alone did not show a predicted interaction with cGAS (S2d Fig). Conversely, the SPRY domains of TRIM16 and TRIM16L share ~97% identity, yet their full-length binding profiles are only weakly correlated; when only the SPRY domains were analyzed, the correlation between their binding profiles increased substantially (S2e Fig). Together, these results suggest that substrate recognition is shaped both by local determinants within the SPRY domain and by the broader structural context of the full-length protein.

To probe this further and assess whether AlphaFold could distinguish substrate specificity among SPRY domains with high sequence similarity, we focused on the E3 ubiquitin ligases TRIM25 and Riplet. These proteins share a common domain architecture, each comprising of a RING domain (which mediates ubiquitin transfer), a coiled-coil region and a C-terminal SPRY domain, which share 30.7% protein identity between the two proteins (Fig 2d). Both ligases have been implicated in modulating the activity of the same nucleic acid sensors, including RIG-I [9,12,13] and ZAP [10,11]. When analyzing interaction scores between all human SPRY-containing proteins and the innate immune sensors ZAP and RIG-I, TRIM25 emerged as the top predicted interactor with ZAP, whereas the Riplet-ZAP interaction scored comparatively low (Fig 2e). Conversely, Riplet was among the top-scoring predicted interactors with RIG-I, while TRIM25 scored poorly in this context. To test if this AlphaFold prediction corroborated what interactions happen in cells, we performed co-immunoprecipitations of ZAP and RIG-I and observed that TRIM25, but not Riplet, co-immunoprecipitated with ZAP, while Riplet, but not TRIM25, interacted with RIG-I (Fig 2f). These data suggest that despite high protein sequence similarity in the SPRY domain, AlphaFold can accurately predict the binding preferences of SPRY-containing proteins in humans.

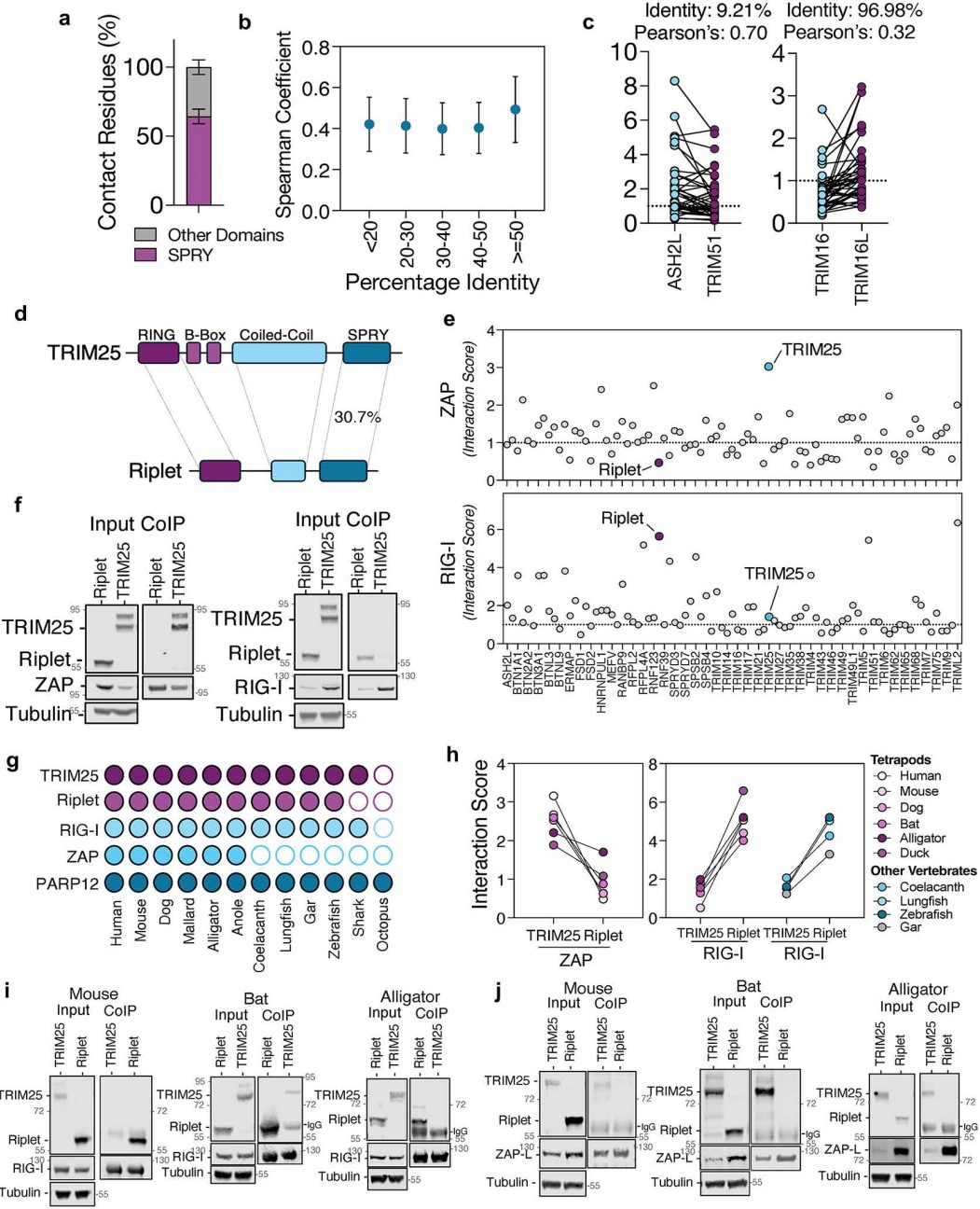

**Fig 2. Sequence similarity does not predict SPRY-sensor interactions.** Proportion of residues located in the SPRY domains of SPRY-containing proteins that interact with nucleic acid sensors (a). Correlation analysis of percent identity of the SPRY domains from all SPRY-containing proteins and their interaction scores across all binding sensors (b). Interaction scores of SPRY-containing proteins with low percent identity but high Pearson's correlation coefficient in their interaction binding profile, and high percent identity but low Pearson's correlation coefficient (c). Connecting lines represent interaction scores between the indicated SPRY proteins and the same "bait" protein. Topological organization of TRIM25 and Riplet (d). Summary of interaction scores for ZAP and RIG-I across all SPRY-containing proteins (e). HEK293T were transfected with plasmids encoding RIG-I-FLAG, ZAP-L-FLAG, TRIM25-HA or Riplet-HA, and cell lysates incubated with anti-FLAG antibodies and analyzed by SDS-PAGE/Western blot (f). Schematic representation of the presence (filled circles) or absence (empty circles) of TRIM25, Riplet, RIG-I, ZAP and PARP12 orthologues in the indicated species (g). Interaction scores between orthologues of ZAP or RIG-I against cognate orthologues of TRIM25 and Riplet (h). HEK293T cells were transfected plasmids encoding with mouse, bat or alligator RIG-I, ZAP, Riplet and TRIM25 - as indicated - and 48h later cells were lysed and RIG-I- (i) or ZAP-complexes (j) were isolated with anti-FLAG antibodies and analyzed by western blotting.

Moreover, previous studies showed that orthologues of TRIM25 in other species can functionally interact with RIG-I [12,22]; thus we hypothesized that the sequence similarity between Riplet and TRIM25 may have resulted in overlapping functions between the two ubiquitin ligases. To understand how these interaction preferences arise, we examined the evolutionary origin of TRIM25 and Riplet by surveying genomic data from a broad range of vertebrate and invertebrate species. TRIM25 orthologs were identified in lineages as ancient as chondrichthyes, including the elephant shark (*Callorhinchus milii*), suggesting that TRIM25 emerged early in vertebrate evolution (Fig 2g). In contrast, Riplet orthologs were restricted to tetrapods, coelacanths and bony fish, but absent from cartilaginous fish, indicating that Riplet likely originated after the divergence of osteichthyes from chondrichthyes. Interestingly, the evolutionary distributions of their respective interaction partners are more limited. While RIG-I is broadly present in vertebrate genomes, ZAP is restricted to tetrapods. This suggests that the TRIM25-ZAP and Riplet-RIG-I interaction axes may have evolved at different evolutionary stages, with ZAP-TRIM25 representing a more recent innovation. We next examined the genomic context of *TRIM25* and *Riplet* across species. In the human genome, both genes reside on chromosome 17, separated by approximately 26 Mb, however, in the genomes of all the other species we analyzed, *TRIM25* and *Riplet* are much closer together, as little as 0.1Mb (S3a Fig). Synteny analysis revealed that this chromosomal co-localization and surrounding gene order are conserved in diverse vertebrate species (S3a Fig). Given this shared evolutionary history, we asked if the recognition of specific substrates by TRIM25 and Riplet is an evolutionary conserved feature. We applied AlphaFold to predict cross-species interaction scores between RIG-I, ZAP, TRIM25 and Riplet (Fig 2h). Across species, RIG-I consistently showed higher predicted affinity for Riplet than for TRIM25, whereas ZAP was predicted to preferentially bind TRIM25 over Riplet. These predictions were experimentally validated via co-immunoprecipitation of orthologous proteins from diverse vertebrates (Fig 2i,2j), confirming the substrate preferences predicted computationally. Overall, these findings indicate that the ability of TRIM25 and Riplet to recognize specific substrates is an evolutionarily conserved trait. The conservation of interaction specificity highlights the robustness of SPRY-mediated substrate recognition across evolutionary timescales.

## SPRY domains dictate substrate specificity

Despite preferential binding, TRIM25 and Riplet may still impact RIG-I- and ZAP-mediated antiviral responses in an interaction-independent manner; thus we investigated the role of each of these SPRY-containing proteins in each sensor's activity. Knockdown of Riplet in human cells significantly impaired transcription of *IFNB1* mRNA in response to 5′ triphosphate hairpin RNA (3p-hRNA) derived from influenza A virus, a canonical RIG-I ligand, whereas TRIM25 knockdown had little effect (Figs 3a, S4a and S4b). This impairment was specific to RIG-I-mediated signaling, as Riplet-depleted cells retained responsiveness to poly(I:C), a ligand for MDA5 and TLR3 (Fig 2b). This effect was observed across multiple human cell lines and extended to both *IFNB1* and *IFNL2* transcripts (S4c, S4d, and S4e Fig). To probe ZAP function, we used a CpG-enriched mutant of enterovirus A71 (EV-A71) that is highly sensitive to ZAP activity (S4f and S4g Fig). TRIM25 depletion rescued replication and viral RNA accumulation of this attenuated virus, whereas Riplet depletion had no effect (Figs 3c and S4h). These results support a model in which Riplet promotes RIG-I-mediated responses while TRIM25 enhances ZAP-mediated antiviral restriction, consistent with AlphaFold predictions of SPRY-specific interactions. Given that SPRY domains are proposed to confer substrate selectivity to E3 ubiquitin ligases, and that TRIM25 and Riplet share conserved domain topology, we asked whether swapping SPRY domains would reprogram substrate recognition. To this end, we generated two chimeric constructs: (1) TRIM25 with the SPRY domain of Riplet (T25-Riplet^SPRY) and (2) Riplet with the SPRY domain of TRIM25 (Riplet-T25^SPRY) (Figs 3d and S5a). AlphaFold predictions indicated that these domain-swapped constructs would enhance interaction with the new target substrates (Fig 3E). Consistent with this prediction, co-immunoprecipitation assays revealed that TRIM25, but not Riplet, interacted with ZAP (Fig 3f). Notably, Riplet-T25^SPRY strongly bound to ZAP, recapitulating TRIM25's interaction profile. Using the T25-Riplet^SPRY instead of the Riplet protein, predicted a higher interaction score despite the lack of specific pull-down between this chimera and ZAP; importantly, this interaction score is below our defined threshold

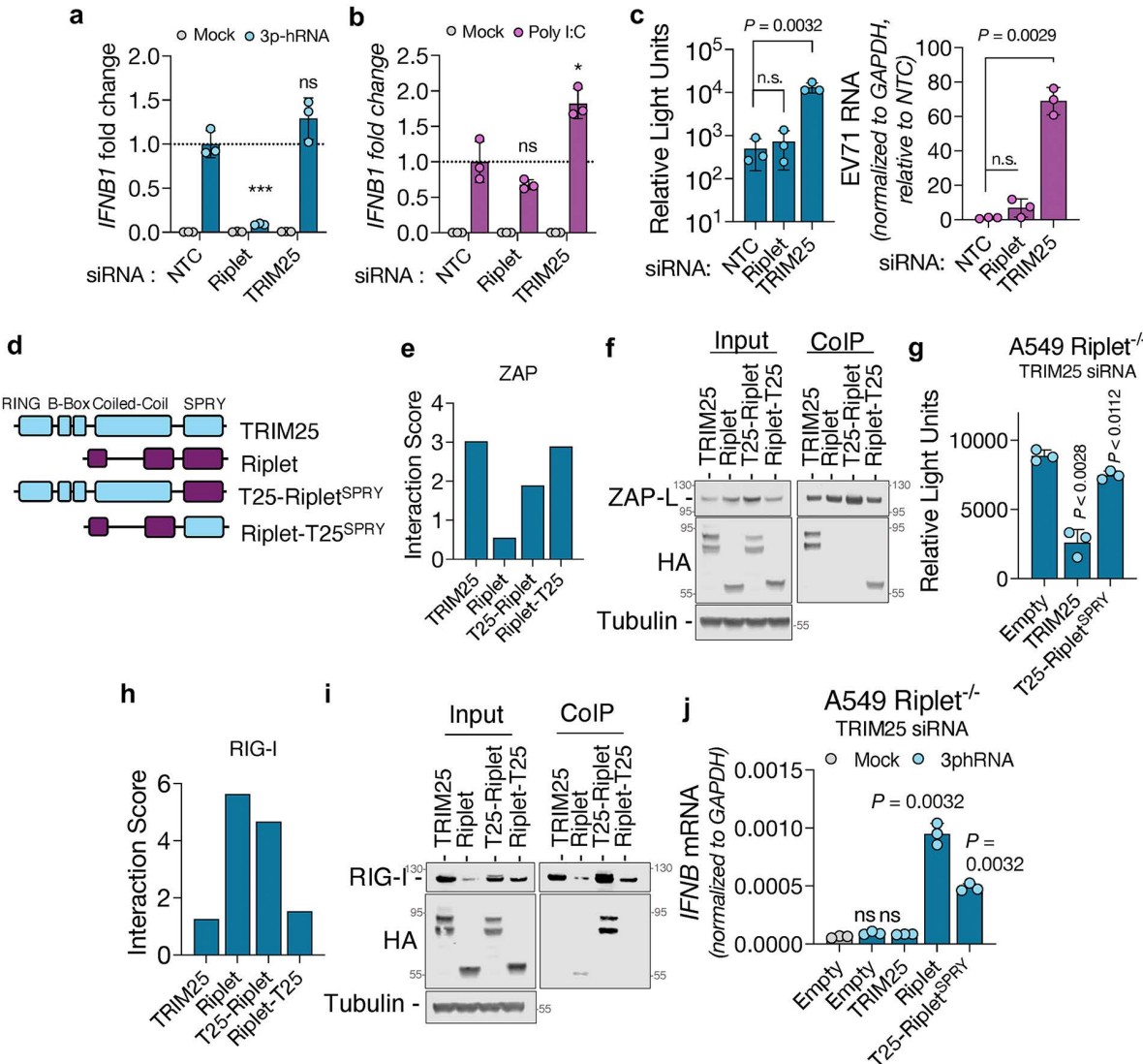

**Fig 3. SPRY domains dictate substrate specificity.** Riplet and TRIM25 were depleted from A549 cells using siRNAs and 48h later cells were transfected with 3p-hRNA (a) or Poly I:C (b). After 8h, RNA was extracted and *IFNB1* transcripts were measured. Riplet- or TRIM25-depleted A549 cells were infected with ZAP-sensitive nLuc-EV71, and 24h later luciferase activity and viral RNA were quantified (c). Schematic representation of the chimeric TRIM25 and Riplet proteins with swapped SPRY domains (d). Interaction scores of chimeric TRIM25 and Riplet proteins against ZAP (e). HEK293T ZAP-/- TRIM25-/- cells were co-transfected with plasmid encoding ZAP-L-FLAG and TRIM25 or Riplet chimeras; protein complexes were isolated from lysates with an anti-FLAG antibody and examined by western blotting (f). A549 Riplet-/- cells reconstituted with an Empty vector or with a vector encoding siRNA-resistant TRIM25 or T25-Riplet^SPRY proteins were transfected with siRNA targeting TRIM25, followed by infection with nLuc-EV71; luciferase activity was measured 24h later (g). Interaction scores of chimeric TRIM25 and Riplet proteins against RIG-I (h). HEK293T cells were co-transfected with plasmid encoding RIG-I-FLAG and TRIM25 or Riplet chimeras; RIG-I was immunoprecipitated and protein complexes were analyzed by western blotting (i). A549 Riplet -/- cells were reconstituted with indicated chimeric proteins; cells were transfected with siRNA targeting TRIM25 and subsequently stimulated with 3p-hRNA (j). *IFNB1* transcripts were measured 8h post-transfection. All data are presented as mean +/- SD (n = 3), *p* values were calculated using unpair Student's t tests against NTC; n.s., not significant.

(2.5), highlighting the importance of benchmarking predicted scores against known interactions. To test the functional consequences of this interaction, we generated a Riplet-knockout A549 cell line (S4i Fig) and used a single siRNA to deplete TRIM25 from these cells. We stability reconstituted these cells by transducing them with retroviruses carrying

CRISPR-resistant Riplet or siRNA-resistant TRIM25 or the respective chimeras. While TRIM25, Riplet and the T25-Riplet<sup>SPRY</sup> chimera had comparable expression (S4j Fig), the Riplet-T25<sup>SPRY</sup> chimera expression very poorly. When cells were infected with the ZAP-sensitive EV-A71 mutant, reconstitution with TRIM25 substantially reduced virus replication while reconstitution with the T25-Riplet<sup>SPRY</sup> chimera only modestly impacted virus replication (Fig 3g), indicating that the SPRY domain of TRIM25 is important to supports ZAP activity. Conversely, we performed similar experiments to assess how these chimeras interact with and support RIG-I's activity. AlphaFold predictions suggested that the T25-Riplet<sup>SPRY</sup> chimera binds to RIG-I, which is evidenced by co-immunoprecipitation assays (Fig 3h and 3i). To determine whether this chimera could functionally substitute endogenous Riplet, we used Riplet<sup>-/-</sup> cells reconstituted with either an empty vector, full-length Riplet or theT25-Riplet<sup>SPRY</sup>. Riplet-knockout cells did not produce IFNB in response to 3p-hRNA stimulation, however, cells reconstituted with Riplet or the T25-Riplet<sup>SPRY</sup> chimera had robust *IFNB1* production (Fig 3j), demonstrating functional replacement of Riplet by the chimeric construct. Taken together, these results indicate that SPRY domains are sufficient to confer substrate specificity to TRIM25 and Riplet, and that AlphaFold can accurately resolve such specificity even among homologous domains. These findings underscore the utility of structure-based prediction in decoding the molecular logic of ubiquitin ligase-substrate recognition.

## Functional divergence of SPRY-containing proteins enables sensor level control

We noticed that overexpression of Riplet led to a decrease in RIG-I expression, while TRIM25 reduced ZAP levels (Fig 2f). Indeed, transfecting increasing amounts of plasmids encoding Riplet led to an incremental decrease in RIG-I (Fig 4a), while increasing amounts of TRIM25 degraded ZAP (S5b Fig). Additionally, reducing Riplet expression in cells increased endogenous levels of RIG-I (Fig 4b), suggesting that RIG-I abundance may be regulated by Riplet. However, the chimera of TRIM25 carrying the SPRY domain of Riplet co-immunoprecipitated with RIG-I and supported RIG-I-mediated IFNB1 production, but did not reduce RIG-I levels (Fig 3i and 3j). These data suggested that the ability of Riplet to promote RIG-I signaling can be genetically separated from its effect on steady-state RIG-I abundance, and further implied that determinants of RIG-I destabilization reside outside the SPRY domain.

To address this, we investigated how Riplet and TRIM25 have diverged in vertebrates (Fig 4c). Phylogenetic analyses of orthologues of TRIM25 and Riplet revealed that while all TRIM25 orthologues and Riplet orthologues from fish, coelacanth and lungfish clustered together, Riplet sequences retrieved from tetrapods formed a separate group, suggesting that Riplet sequences found in non-tetrapods are more similar to TRIM25 than Riplet sequences from tetrapods. We then used AlphaFold3 to generate predicted structures of TRIM25/Riplet orthologs across multiple species and compared them to their human counterparts. All orthologs retained a conserved domain organization with an N-terminal RING domain and a C-terminal SPRY domain. However, the intervening regions exhibited substantial diversity. TRIM25 orthologs consistently possessed two N-proximal B-box domains and a long alpha-helical region forming a coiled-coil (H1), followed by two shorter helices (H2 and H3), as previously described [23], preserving the canonical topology across species. Riplet orthologs, by contrast, showed marked structural divergence. Mammalian Riplet proteins lacked B-box domains and exhibited a severely truncated coiled-coil domain. In birds and reptiles, B-box domains remained absent, but the coiled-coil regions more closely resembled those of TRIM25 (Fig 4c and 4d). In contrast, Riplet proteins from lungfish, coelacanth and fish retained B-box domains and extended coiled-coil regions, resembling TRIM25-like topology. These observations suggest that TRIM25 and Riplet evolved from a common ancestral gene, plausibly a TRIM25-like protein, with Riplet undergoing extensive domain loss and rearrangement in higher vertebrates. The most prominent change in mammalian Riplet is the shortening of the coiled-coil domain, which is known in TRIM25 to mediate dimerization [23]. We hypothesized that truncation of this domain may underlie a functional shift from a dimeric to a monomeric state. To test this, we performed in-cell glutaraldehyde crosslinking assays to assess oligomeric status (Fig 4e). Crosslinked human TRIM25 migrated as a dimer, whereas Riplet appeared exclusively monomeric. These findings were consistent with the predicted structural differences in the coiled-coil domains.

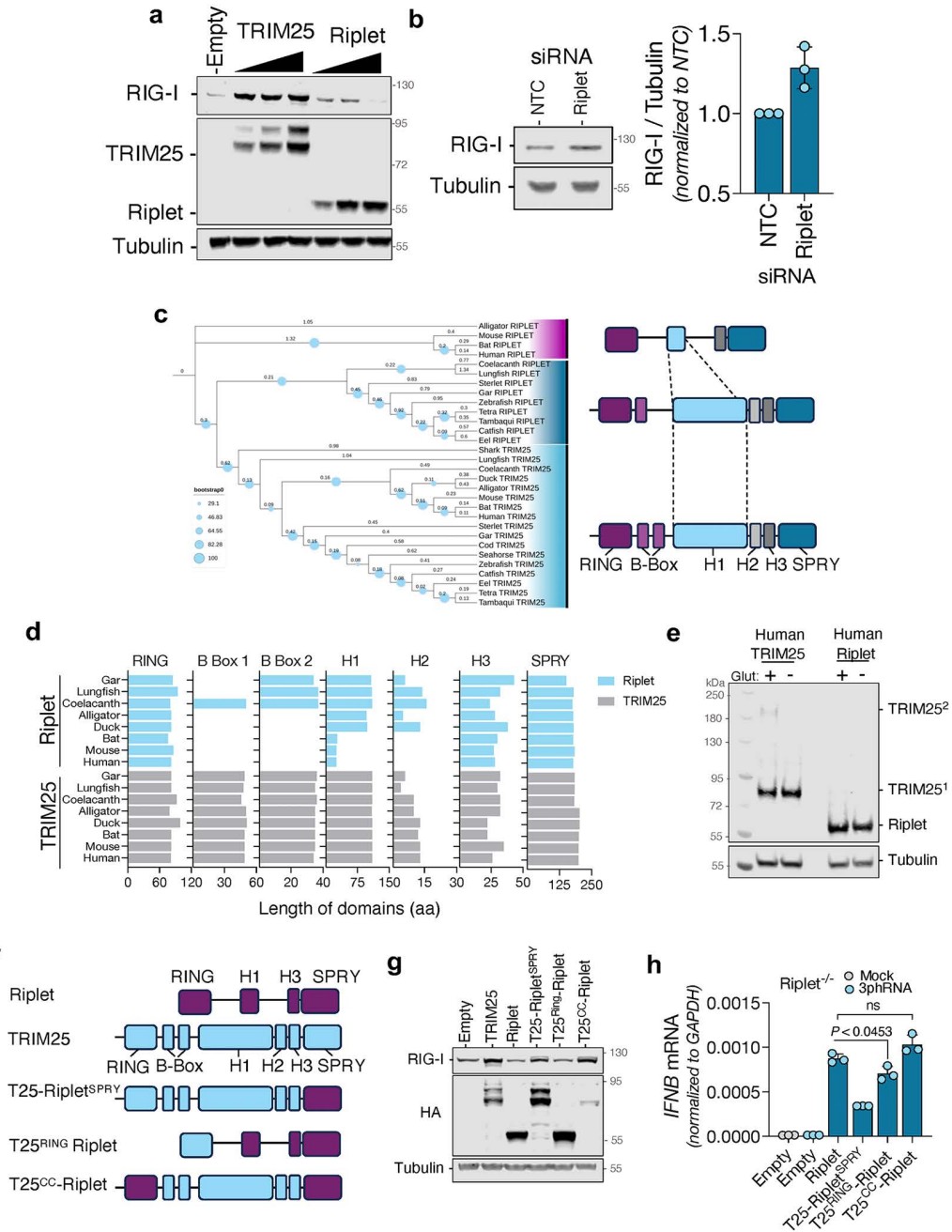

**Fig 4. Functional divergence of SPRY-containing proteins enables sensor level control.** HEK293T RIG-I$^{-/-}$ cells were transfected with increasing amounts of plasmids encoding TRIM25-HA or Riplet-HA and RIG-I expression was analyzed by western blotting (a). A549 cells were transfected with siRNAs targeting Riplet and 72h post-transfection, RIG-I levels were measured by western blotting (b). Maximum-likelihood tree (midpoint rooted) showing phylogenetic relationships among TRIM25 and Riplet-related protein sequences in vertebrates generated with iqTREE and iTOL with 1000 bootstrap replicates. Topological representation of TRIM25/Riplet sequences in species shaded in blue and pink (c). Protein length of RING, B Box, SPRY domains as well as helices 1, 2 and 3 - as described in Sanchez et al. 2014 [23] - of TRIM25 and Riplet orthologues (d). Cells expressing human TRIM25 and Riplet were treated with glutaraldehyde and analyzed by western blotting (e). Schematic representation of Riplet-TRIM25 chimeras used: one Riplet chimera in which the RING domain swapped by the RING domain of TRIM25 (T25$^{RING}$-Riplet) and a chimera carrying the RING and SPRY domains of Riplet and the B-box and coiled-coil domain of TRIM25 (Riplet$^{RING}$ T25$^{CC}$-Riplet$^{SPRY}$) (f). HEK293T cells were transfected with plasmid encoding chimeric TRIM25 and Riplet proteins, and RIG-I expression levels was analyzed by western blotting (g). A549 Riplet$^{-/-}$ reconstituted with vectors encoding the indicated proteins were stimulated with 3p-hRNA and 8h later *IFNB1* transcripts were measured by qPCR. Data is representative of three independent replicates.

To test whether these architectural differences influence Riplet-dependent control of RIG-I, we generated two additional chimeras: one in which the RING domain of Riplet was replaced by the RING domain of TRIM25 (T25^RING-Riplet), and another containing the RING and SPRY domains of Riplet together with the long coiled-coil region of TRIM25 (T25^CC-Riplet) (Fig 4f). Replacement of the Riplet RING domain had little effect on the ability of Riplet to reduce RIG-I abundance, indicating that the RING domains of the two proteins are functionally interchangeable in this assay (Fig 4g). By contrast, insertion of the TRIM25 coiled-coil region into Riplet abolished the reduction in RIG-I levels, indicating that the truncated Riplet architecture, and particularly its native coiled-coil configuration, contributes to destabilization of the sensor (Fig 4g). Importantly, these effects on RIG-I abundance did not track directly with signaling output. In the IFNB1 induction assay, the T25^RING-Riplet chimera remained strongly active, and the T25^CC-Riplet chimera retained signaling activity despite failing to reduce RIG-I abundance (Fig 4h). Together, these results indicate that Riplet-dependent control of RIG-I protein abundance and Riplet-dependent activation of downstream signaling are separable outputs with distinct domain requirements: the SPRY domain is sufficient to confer substantial signaling competence, whereas the native Riplet architecture, particularly its shortened coiled-coil region, is required for efficient reduction of RIG-I levels. Thus, while Riplet regulate RIG-I levels in resting cells a manner that is dependent on its coiled-coil domain, RIG-I degradation upon activation is dispensable for IFN production.

## Residues in the variable loops of SPRY mediate substrate specificity

We then probed the molecular basis of specificity by mapping residue-level contacts in SPRY loops. To investigate how SPRY domains achieve substrate specificity, we analyzed high-confidence interactions between nucleic acid sensors and SPRY-containing E3 ubiquitin ligases. Structural predictions revealed that the majority of SPRY-substrate contacts are mediated by residues within five hypervariable loops of the SPRY domain (Figs 5a and S6a), which are known to contribute to target recognition in several TRIM family members. Consistently, AlphaFold3 accurately predicted residue-level contacts for protein pairs with known interactions, including Riplet-RIG-I and SPSB3-cGAS. We next evaluated whether AlphaFold3 could resolve the molecular interface of protein complexes lacking high-resolution structural data. To this end, we focused on the TRIM25-ZAP interaction, a well-documented functional interaction for which the precise molecular contacts remain undefined. Structure prediction indicated that ZAP interacts with TRIM25 through its SPRY domain, in agreement with prior co-immunoprecipitation studies [10,24–26] The interface included four predicted salt bridges between the side chains of the following amino acids: D3, E5, E39 and K122 of ZAP formed electrostatic interactions with R541, R544, E539 and K469 of TRIM25, respectively (Fig 5b). Additionally, the residue Q36 of ZAP was predicted to interact with the backbone of F618 of TRIM25. These charged residues are conserved across tetrapod orthologs (S6b Fig), suggesting evolutionary constraint on this interaction surface. To experimentally assess the functional relevance of these residues, we engineered a ZAP quadruple mutant (ZAP^qMut), in which D3, E5 and E39 were substituted with arginine and K122 with glutamate (S6c Fig). Co-immunoprecipitation assays revealed that this mutant exhibited markedly reduced binding to TRIM25 compared to wild-type ZAP, a feature that was observed in both long and short isoforms of ZAP (Figs 5c and S6d). To determine whether impaired interaction with TRIM25 compromised antiviral function, we reconstituted ZAP-knockout cells with either ZAP^WT or ZAP^qMut and infected them with ZAP-sensitive enterovirus A71. Cells expressing ZAP^qMut failed to restrict viral replication, demonstrating that the TRIM25-ZAP interaction is critical for antiviral activity (Figs 5d and S6e). Given the electrostatic nature of the predicted interface, we asked whether restoring charge complementarity in TRIM25 could rescue interaction with ZAP^qMut. While most charge-reversal mutations in TRIM25 (R541D, R544E, E539K or the quadruple mutant) impaired protein stability (S6f Fig), a K469E mutant was expressed at detectable levels. Remarkably, although the TRIM25^K469E mutant exhibited reduced binding to ZAP^WT, co-expression with the ZAP^E39K variant partially restored binding (Figs 5e and S6g), supporting the functional relevance of this specific salt bridge. To rule out the possibility that the ZAP^qMut substitutions globally

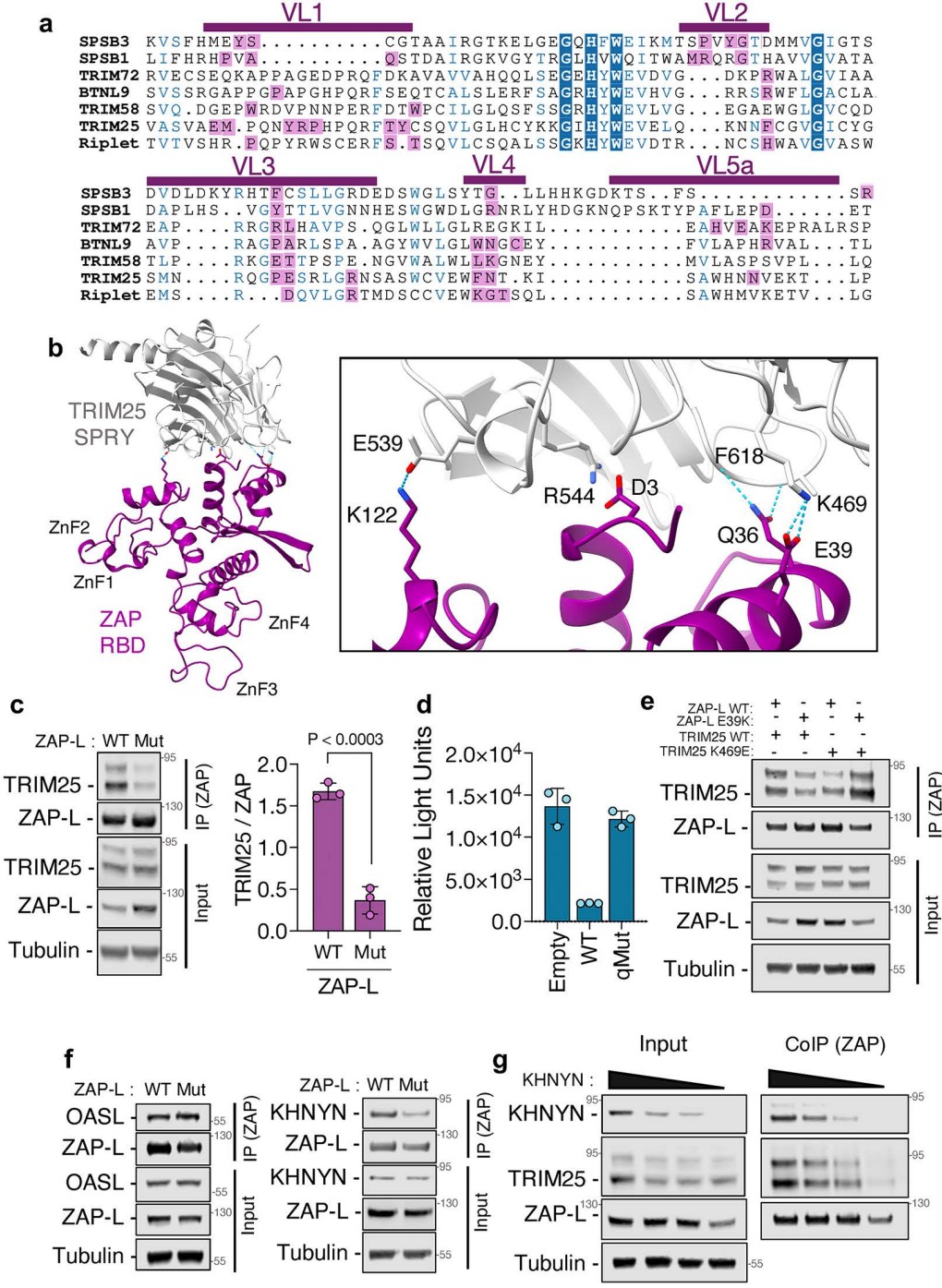

**Fig 5. Residues in the variable loops of SPRY interact with substrates.** Multiple sequence alignment of the SPRY domains of SPSB3, TRIM72, BTNL9, TRIM58, TRIM25 and Riplet. Residues interacting with cognate sensors are indicated in purple (a). Variable loops (VL) are annotated as in James et al. 2007 [17]. Structure prediction of the RNA-binding domain (RBD) of ZAP (purple) and the SPRY domain of TRIM25 (grey) with contact residues highlighted (b). Hydrogen bonds are shown in dashed lines. HEK293T ZAP[-/-] TRIM25[-/-] cells were transfected with plasmids encoding TRIM25 and either wildtype (WT) or a mutant of ZAP-L carrying D3K, E5K, E39K and K122E substitution (Mut), and 48h later ZAP was isolated from cell lysates and analyzed by western blotting (c). Quantification of pulldown TRIM25 is illustrated in the purple bars. HeLa ZAP[-/-] cells were reconstituted with an empty vector, or vectors encoding wildtype or mutant ZAP and infected with nLuc-EV71. Luciferase activity was measured 24h later (d). HEK293T ZAP[-/-] TRIM25[-/-] cells were transfected with plasmid encoding the indicated wildtype and mutant versions of ZAP and TRIM25, and ZAP complexes were

isolated from cell lysates 48h later (e). HEK293T ZAP$^{-/-}$ TRIM25$^{-/-}$ KHNYN$^{-/-}$ N4BP1$^{-/-}$ cells were transfected with plasmid encoding ZAP-L wildtype (WT) or Mut (D3K, E5K, E39K and K122E) along with OASL or KHNYN. ZAP-bound complexes were isolated from cell lysates and resolved by SDS-PAGE/western blotting (f). HEK293T ZAP$^{-/-}$ TRIM25$^{-/-}$ KHNYN$^{-/-}$ N4BP1$^{-/-}$ cells were transfected with 500ng of plasmids encoding ZAP-L and TRIM25, as well as increasing amounts of a plasmid encoding a catalytically inactive HA-tagged KHNYN. ZAP-complexes were purified from cell lysates and analyzed by western blotting. All data are presented as mean +/- SD (n = 3), p values were calculated using unpaired Student's t tests.

disrupted protein folding, we tested its ability to interact with two unrelated ZAP-binding partners: OASL and KHNYN [8,25,27]. ZAP$^{WT}$ and ZAP$^{qMut}$ interacted equally well with OASL, but ZAP$^{qMut}$ showed significantly reduced binding to KHNYN (Figs 5f and S6h-S6j). Structure predictions indicated that OASL binds to the N-terminal RNA-binding domain of ZAP, whereas KHNYN occupies a region overlapping the TRIM25 interface (S6k-S6l Fig). This is consistent with crystallographic data of the ZAP-KHNYN complex [25,28,29], suggesting that both KHNYN and TRIM25 engage a common binding site on ZAP. Given that both TRIM25 and KHNYN are required for ZAP-mediated antiviral function, we hypothesized they may compete for binding to ZAP. To test this, we performed co-immunoprecipitation experiments in which TRIM25 and ZAP$^{WT}$ levels were held constant while increasing amounts of KHNYN were expressed. Surprisingly, we observed a proportional increase in TRIM25 co-immunoprecipitation with ZAP in response to KHNYN overexpression (Fig 5g), indicating a cooperative, rather than competitive, binding mode. Together, these results demonstrate that AlphaFold3 can accurately predict residue-specific contacts in protein-protein interactions, even in the absence of prior structural data. Moreover, the functional importance of these contacts is underscored by mutational analyses, providing a mechanistic explanation for substrate recognition by SPRY domains.

## Incoming RNA is surveilled by different SPRY-containing proteins

Both TRIM25 and Riplet play essential roles in the cellular response to RNA [30]. To examine their contribution in a physiologically relevant context, we turned to self-amplifying RNA (saRNA) vaccine platforms [31]. Our saRNA is derived from an alphavirus replicon (Venezuelan equine encephalitis virus, VEEV) that encodes the viral replication machinery but lacks structural genes, rendering it replication-competent but non-infectious [32]. This platform enables antigen expression at significantly lower RNA doses compared to traditional mRNA vaccines [33]. Although saRNA shows strong immunogenicity in small animal models, its efficacy in humans remains suboptimal [34]. Given that both ZAP and RIG-I detect alphavirus RNA during natural infection [35,36], we hypothesized that their associated SPRY-containing adaptors (TRIM25 and Riplet) might modulate the innate immune response to saRNA, potentially limiting its effectiveness in human applications. To determine the functional roles of TRIM25 and Riplet in modulating saRNA activity, we depleted each protein via siRNA and transfected cells with a firefly luciferase-encoding saRNA construct, delivering saRNA either via lipofectamine (a lipoplex formulation with permanently cationic lipids) or a lipid nanoparticle (LNP) formulation with ionizable cationic lipids similar to those in clinically approved RNA vaccines [37]. Notably, Riplet depletion significantly enhanced saRNA-driven luciferase expression in the context of lipofectamine delivery but had no effect under LNP conditions (Fig 6a). In contrast, TRIM25 depletion increased saRNA activity following LNP delivery but not lipofection (Fig 6a). Since Riplet supports RIG-I activity, which initiates a signaling cascade that culminates in the production of interferons, we assessed the ability of each delivery method to trigger type I interferon responses. saRNA introduced by either lipofectamine or LNP stimulated IFNB1 transcription, but lipofectamine elicited a markedly stronger response (Fig 6b and 6c). Correspondingly, expression of interferon-stimulated genes, such as TRIM25 and the short isoform of ZAP (ZAP-S), was robustly upregulated in lipofectamine-transfected cells but remained low following LNP delivery (Fig 6d). Next, we evaluated the contribution of RIG-I to this immune response and subsequent restriction of saRNA-derived luciferase expression. RIG-I depletion enhanced saRNA activity only when delivered via lipofectamine (Fig 6e), consistent with its known activation via cytoplasmic recognition of RNA. Furthermore, RIG-I or Riplet knockdown suppressed IFNB1 production in lipofectamine-transfected cells, confirming their role in type I IFN induction under these conditions (Fig 6f). These data suggest that

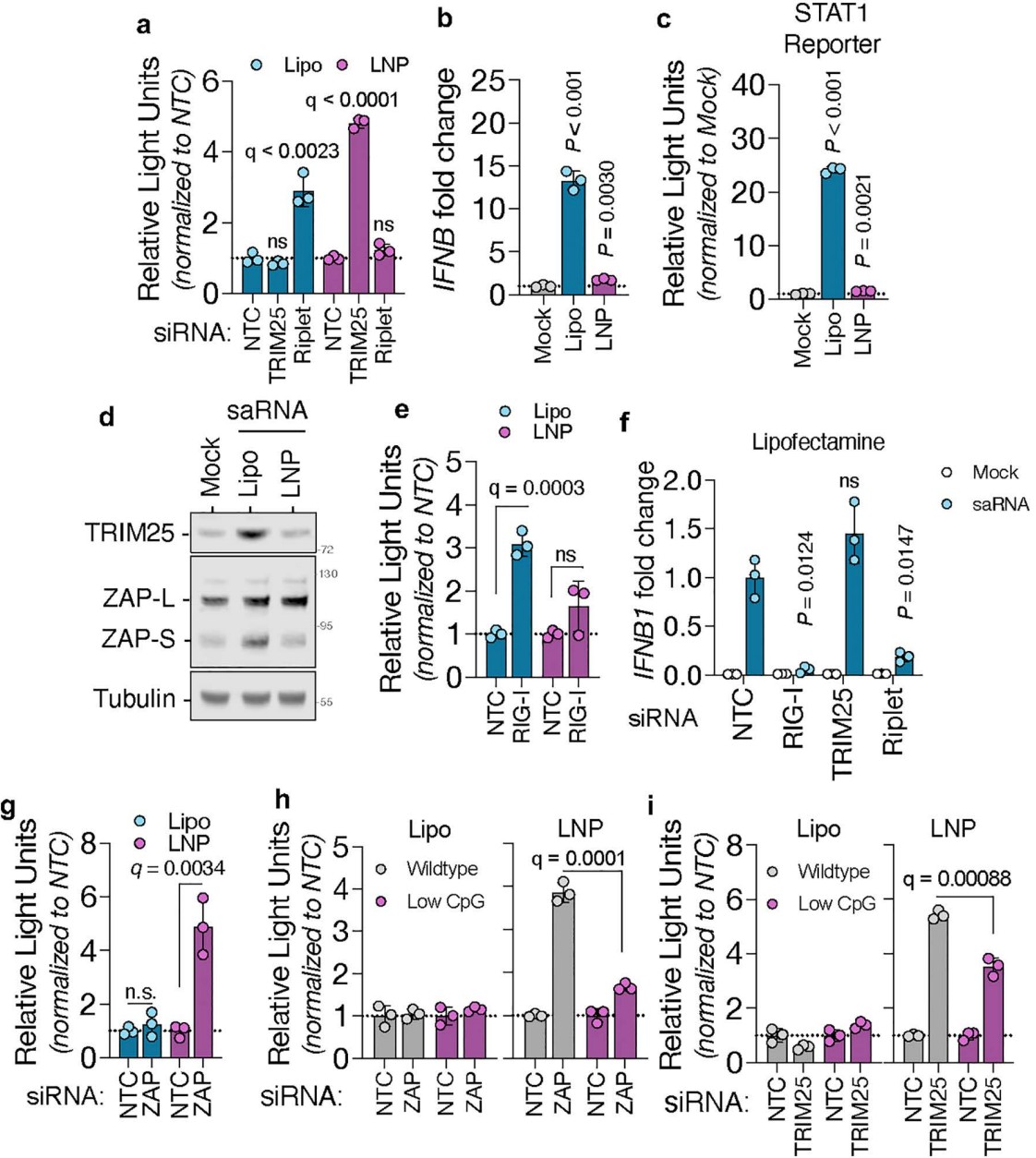

**Fig 6. Incoming RNA is surveilled by different SPRY-containing proteins.** TRIM25 or Riplet were depleted from A549 cells using siRNAs and 48h later cells were transfected with self-amplifying RNA (saRNA), carrying a luciferase reporter, formulated in a lipid nanoparticle (LNP) or by using lipofectamine (Lipo) (a). A549 cells were transfected with saRNA by LNP or lipofectamine and 8h later total RNA was extracted and *IFNB* transcripts were measured by qPCR (b). A549 cells containing a reporter firefly luciferase gene under the control of the IRF9/STAT1 promoter were transfected with saRNA by LNP or lipofectamine; luciferase activity (c) and the expression of TRIM25 and ZAP (d) were assessed 24h post-transfection. RIG-I was depleted from A549 cells using siRNAs and 48h later cells were transfected with LNP- or Lipofectamine-saRNA complexes (e). Luciferase activity was assessed 24h post-transfection. A549 cells were transfected with siRNA targeting RIG-I, TRIM25 or Riplet and subsequently transfected with saRNA in complex with lipofectamine. Production of *IFNB* transcripts was assessed 8h after transfection (f). ZAP was depleted from A549 cells and, 48h later, cells were transfected with LNP- or Lipofectamine-saRNA complexes (g). Luciferase activity was measured 24h later. ZAP (h) or TRIM25 and Riplet (i) were depleted from A549 cells using siRNAs and 48h later cells were transfected with either wildtype saRNA (containing 341 CpGs) or a synonymously recoded, CpG-low saRNA (containing 15 CpGs) formulated in LNP or by lipofectamine. Luciferase activity was measured 24h after transfection. All data are presented as mean +/- SD (n = 3), *q*-values were calculated using multiple unpaired t tests against NTC; ns, not significant.

saRNA encapsulated in LNPs engages the innate immune system less potently than lipofectamine-transfected saRNA, and this immune response is mediated by RIG-I and Riplet but not TRIM25. Since TRIM25-depletion increases luciferase activity in cells transfected with LNP-formulated saRNA, we next asked whether ZAP mediates restriction of saRNA delivered by LNP. ZAP knockdown had no effect on lipofectamine-mediated transfection, but significantly enhanced luciferase activity from saRNA delivered via LNP (Fig 6g). These data suggest that ZAP, likely in cooperation with TRIM25, restricts LNP-delivered saRNA. ZAP is known to recognize CpG dinucleotides within viral RNA [7,38], so we investigated whether removing CpGs from the saRNA construct could mitigate ZAP-mediated restriction. A CpG-depleted version of saRNA exhibited markedly reduced dependence on ZAP following LNP transfection (Fig 6h). Furthermore, this construct also showed diminished sensitivity to TRIM25 depletion (Fig 6i), suggesting that TRIM25-dependent restriction is, at least in part, CpG-dependent. Collectively, these findings demonstrate that TRIM25 and Riplet regulate saRNA activity in a delivery-specific manner, with TRIM25-ZAP preferentially restricting LNP-encapsulated RNA and Riplet-RIG-I acting primarily during lipofection. These insights have important implications for optimizing saRNA vaccine design and delivery in humans.

## Discussion

Detection of viral RNA is central to initiating protective innate immunity, but dysregulation of RNA sensors can amplify inflammatory responses and contribute to autoimmunity [39]. Ubiquitin ligases have emerged as key regulators of this process, functioning both as activators and as brakes on RNA sensor signaling. Yet, because many of these interactions are transient or degradation-sensitive, they are often invisible to conventional high-throughput methods such as affinity purification coupled with mass spectrometry. To address this challenge, we applied AlphaFold-based structural prediction to systematically identify novel interactions between RNA sensors and ubiquitin ligases. While AlphaFold has been widely used to model protein structure, its use in unbiased discovery of protein-protein interactions remains limited. Previous efforts such as AlphaPulldown employed AlphaFold-Multimer to interrogate small protein subsets [40] and more recent studies have applied similar strategies to design inhibitory peptides [41]. To our knowledge, our work represents the first large-scale structural screen of immune-relevant protein-protein interactions. This approach uncovered numerous novel interactions validated by biochemical assays, including OAS2-Riplet, STING-BNTL9 and OAS1-TRIM58 (Fig 1d-1g). TRIM58, an E3 ligase implicated in cancer [42,43], has been reported to target proteins such as TLR2 and ZEB1 for proteasomal degradation [44,45]. We found that TRIM58 also downregulates OAS1, an antiviral enzyme that generates 2'-5'-oligoadenylates to activate RNase L. Wildtype TRIM58 reduced OAS1 abundance, whereas a catalytically impaired mutant failed to do so (Fig 1g). These results suggest that TRIM58 limits OAS1 levels and thereby modulates RNase L activity. Given that dysregulated 2'-5'-oligoadenylate production has been linked to lymphoma [46] and that excessive RNase L activation promotes cell death [47], TRIM58 may act as a safeguard to prevent pathological RNA decay. Additional research is needed to determine the role of TRIM58 in limiting OAS1-RNase L activity during immune activation. While our screen revealed novel interactions between SPRY domain containing proteins and innate immune sensors, it is plausible that certain interactions were missed. Interactions that rely on a small number of contacting residues or interactions that may be mediated by a third binding partner are likely to score poorly in our approach. Similarly, the oligomeric nature of some SPRY-containing proteins [23] is likely to impact the correct placement of these domains and, therefore, impact the predicted interaction score. Future optimization of our algorithm will improve the detection of such interactions.

More broadly, our results argue against a simple homology-based model for SPRY-mediated target recognition. While the SPRY domain forms the dominant interaction interface in highly scored complexes, the relationship between SPRY sequence identity and predicted target overlap was weak across the family. In addition, comparison of SPRY-only and full-length interaction screens showed only moderate concordance, indicating that the isolated SPRY domain captures an important component of specificity, but that the surrounding protein context can substantially modulate predicted binding.

These findings provide a plausible explanation for the seemingly paradoxical examples in Fig 2c: distantly related SPRY proteins may converge on the same targets if they retain similar local binding determinants, whereas nearly identical SPRY domains may diverge in target preference if small local substitutions or contacts contributed by non-SPRY regions reshape the effective interaction interface.

Our structural screen also clarified the substrate preferences of Riplet and TRIM25. Across predictions, co-immunoprecipitation and functional assays (Figs 2f-2h and 3a-3c), RIG-I preferentially interacted with Riplet, whereas ZAP engaged TRIM25. While TRIM25 has long been proposed to ubiquitinate RIG-I [12,48–50], more recent studies support our observation that Riplet is the primary ligase acting on RIG-I at physiological concentrations [9,51,52]. One explanation for this discrepancy is that TRIM25, being interferon-inducible, may only reach sufficient levels to bind RIG-I following prolonged signaling. Indeed, purified TRIM25 can ubiquitinate RIG-I in vitro, albeit at much higher concentration than purified Riplet [9]. However, our data suggests that depletion of endogenous TRIM25 slightly augments *IFNB* production (Figs 3a and 6f). This effect may not result from a direct TRIM25-RIG-I interaction, but instead from TRIM25's role in supporting ZAP-mediated degradation of specific endogenous transcripts, particularly interferon mRNAs. Recent studies have demonstrated that abrogation of ZAP expression in human cells and in mice can extend the stability of IFN mRNA transcripts after immune stimulation [53,54]. The functional dichotomy between Riplet and TRIM25 became particularly evident in the context of saRNA vaccines. Lipofectamine-delivered saRNA was exclusively restricted by the Riplet-RIG-I axis, while LNP-delivered saRNA was sensitive to TRIM25-ZAP (Fig 6). One possible explanation is that RIG-I activation requires cytoskeletal remodeling and relocalization of PPP1R12C [55]. Although both lipofectamine and LNP delivery can perturb cortical actin [56,57], whether PPP1R12C is differentially mobilized by these two methods remains unknown. Alternatively, the stage of endosomal escape may dictate sensor engagement. Some LNPs have been shown to release RNA predominantly from late endosomes [58], whereas early or recycling endosomes can also support escape of nucleic acids [59]. Since Riplet localizes to intracellular vesicles [60] and RIG-I can relocalize to early endosomes upon immune sensing [61], differential endosomal routing of lipofectamine versus LNP cargo may underlie their distinct immune signatures. In contrast, TRIM25 and ZAP were exclusively active against LNP-delivered RNA. The long isoform of ZAP, ZAP-L, has been shown to be associated with intracellular vesicles via a C-terminal S-farnesylated residue [62] and Rab11a+ and Rab25 + endosomes [63], postulating a role of ZAP in surveilling the vesicular network. Furthermore, a recent work shows that TRIM25 is a major barrier to LNP-formulated mRNA, acting through pH-dependent binding to endosome-released transcripts [64]. Our data extend this by showing that CpG depletion abolishes ZAP recognition and strongly reduces TRIM25 activity (Fig 6h and 6i), indicating that TRIM25 restricts incoming RNA via two separate mechanisms: (1) by supporting ZAP-mediated RNA recognition and (2) by directly binding to RNA. Indeed, TRIM25 has been shown to bind viral RNAs independently of ZAP, restricting viruses otherwise resistant to ZAP activity [65–67]. While previous studies have demonstrated that the addition of CpG dinucleotides is an effective strategy for the generation of live-attenuated vaccines [68–70], this is, to our knowledge, the first demonstration that reduction of CpG frequency can improve RNA vaccine antigen expression. Together, these findings identify TRIM25 as a central sensor of LNP cargo and suggest that modulating its binding preferences could improve RNA therapeutic delivery while minimizing innate immune restriction.

TRIM25 has been shown to support ZAP's antiviral activity in many virus infections, including sindbis virus [10], murine leukemia virus [26], HIV-1 [24] and ebola virus [71]. Yet, the mechanistic basis of this cooperation has remained unclear. Prior studies demonstrated that TRIM25 interacts with the RNA-binding domain of ZAP via its SPRY domain [10,24,25], but the residues involved in this interaction were still unknown. Our protein-protein interaction prediction pinpoint key residues in the N-terminus of ZAP (D3, E5, Q36, E39, K122) that interact with the SPRY domain of TRIM25, corroborating the importance of these two domains in the interaction. Indeed, early mutagenesis studies of ZAP indicated that residues present in the N-terminus of ZAP abolished antiviral activity [72], however the underlying mechanism was not known. By swapping the amino acids with residues with opposite charge, we observed that TRIM25-ZAP

interaction is almost completely abolished (Fig 5c), but it can be restored with complementary mutagenesis of the SPRY domain of TRIM25 (Fig 5e). Importantly, abrogation of TRIM25-ZAP interaction also led to a reduction in antiviral activity, underlying the importance of these residues in controlling virus infection (Fig 5d). Surprisingly, we found that amino acid substitutions also impacted KHNYN-ZAP interaction. Human KHNYN is composed of an N-terminal KH-homology (KH) domain, a central NYN (N4BP1, YacP-like Nuclease) domain and a C-terminal domain (CTD). The CTD of KHNYN has been shown to interact with ZAP [25] while the NYN domain binds to and degrades RNA in vitro [28]. Interestingly, the KH domain does not bind RNA, suggesting that substrate specificity is mediated by ZAP [28,29]. Our predictions showed that while ZAP-KHNYN and ZAP-TRIM25 binding interfaces overlap, binding of TRIM25 or KHNYN to ZAP is not competitive and instead increasing concentrations of KHNYN stabilize ZAP-TRIM25 interaction (Fig 5g). A recent study shows that ZAP-TRIM25-KHNYN complexes are resistant to RNase treatment and such complexes are stabilized during virus infection [73]. Since ZAP-TRIM25, ZAP-KHNYN and TRIM25-KHNYN interactions can occur on their own [8], we propose that TRIM25 and KHNYN recognize partially overlapping interfaces on ZAP, with each interaction being individually weak or transient. Cooperative binding may arise through local conformational stabilization, oligomerization or multivalent contacts. Determining the stoichiometry and assembly dynamics of this tripartite complex will be an important direction for future work.

In summary, our work establishes SPRY domains as modular determinants of ubiquitin ligase specificity, resolving how these proteins diversify antiviral recognition across evolution while maintaining conserved sensor preferences. By integrating AlphaFold-based structural predictions with experimental validation, we identified both stable and transient ligase-sensor complexes, including interactions inaccessible to conventional proteomics. These insights not only clarify long-standing questions in innate immune regulation but also reveal practical avenues for tuning RNA vaccine performance through manipulation of SPRY-mediated surveillance. More broadly, this framework provides a blueprint for systematically mapping ubiquitin networks and for designing strategies to modulate immune responses in infection, autoimmunity and therapeutic contexts.

## Methods

### Cells

Human embryonic kidney 293T (HEK293T) cells, adenocarcinoma A549 cells, HeLa cells and rhabdomyosarcoma (RD) cells were cultured in Dulbecco's modified medium (DMEM) complemented with fetal bovine serum (10%) and gentamycin. THP-1 cells were cultured in RPMI medium supplemented with fetal bovine serum (10%) and gentamycin. All cell lines were maintained at 37°C and 5% $CO_2$. HEK293T RIG-I$^{-/-}$ were generated by transduction with the lentiCRISPRv2 vector containing a guide RNA targeting exon 1 of *RIGI* (5'- GCATGACCACCGAGCAGCGA3') followed by selection in puromycin. Single-cell clones were derived by limiting dilution and loss of RIG-I protein expression and DNA lesion was confirmed by western blotting and by sequencing the genomic locus. A549 Riplet$^{-/-}$ cells were generated by transduction with the lentiCRISPRv2 vector containing a guide RNA targeting exon 1 of *RNF135* (5'-CACGCGAGATACAGGCGGGC-3'), followed by single cell clone isolation. DNA lesion at the exon 1 locus was confirmed by DNA sequencing.

### Plasmids

Plasmid encoding human FLAG-tagged ZAP-L, human HA-tagged TRIM25 and human HA-tagged TRIM58 were used to introduce point mutations at indicated residues by site-directed mutagenesis (KLD Enzyme Mix, NEB) according to the manufacturer's guidelines. To clone human BTNL9, OAS2 and SPSB1 expression constructs, total RNA was extracted from A549 cells using the NucleoSpin RNA, Mini kit for RNA purification (Macherey-Nagel) according to the guidelines provided by the manufacturer. RNA was then reverse transcribed using the Superscript IV First Strand Synthesis kit

(Invitrogen) using random hexamers and the *BTNL9* and *OAS2* open reading frames were amplified by PCR and cloned into the pCR3.1. Human *OAS1* and *STING* open reading frames were cloned from pLX302 OAS1-V5 puro ([74]Addgene #158641) and pMRX-hSTING-EGFP ([75], Addene #214149), respectively, amplified by PCR and cloned into pCR3.1 plasmid carrying a Flag epitope. The sequences of human TRIM58 (CCDS1636.1), *Eptesicus fuscus* (Big brown bat) RIG-I (XM_008142005.3) and Riplet (XM_054709681.1), *Alligator mississippiensis* RIG-I (XM_019490707.2) and Riplet (XM_014601844.3) were retrieved from NCBI CCDS or Nucleotide databases, synthesized by Twist Biosciences and cloned into pCR3.1 expression constructs. Mouse (*Mus musculus*) RIG-I and Riplet open reading frames were amplified from cDNA generated from total RNA extracts of BALB/C mice lung samples [76]. Plasmids encoding ZAP-L and TRIM25 of human, mouse, bat and alligator origin have been previously described [24]. Chimeric versions of TRIM25 and Riplet were performed using the following domain boundaries: TRIM25 SPRY 404–630aa, Riplet SPRY 241–432aa, TRIM25 RING 1–86aa and Riplet RING 1–97aa.

### Viruses

Recombinant Enterovirus A71 (strain 41) carrying the nano luciferase gene and ZAP-sensitive synonymous muta-tions (CG48/A$^+$) [68] were recovered by in intro RNA transcription and transfection as previously described [77]. Two days after transfection, supernatants were collected, clarified by centrifugation and used to inoculate RD cells. Cells were monitored for extensive cytopathic effect, supernatants were then collected, clarified by centrifugation at 2,000xg for 10min at 4˚C and filtered through a 0.22µm filter. Virus stocks (P1) were stored at -70˚C until fur-ther use. For virus infections, HeLa or A549 cells were seeded onto 12-well plates; after 24h, supernatants were removed and cells were incubated with 500µL of inoculum at a multiplicity of infection (MOI) of 0.02 in serum-free DMEM for 1h at 37˚C. Cells were then washed thrice in PBS and incubated at 37˚C in complete DMEM for the indicated times. Cells were then lysed in RA1 buffer for RNA extraction using the Nucleospin RNA kit (Macherey-Nagel), or in Passive Lysis Buffer (Promega) for nano-luciferase activity measurement. Isolated RNA was reverse transcribed using the Superscript IV First Strand Synthesis kit (Invitrogen), and viral RNA was quantified as previ-ously described [68].

### Transduction of cell lines

To reconstitute ZAP$^{-/-}$ cells, a mutant of ZAP-L carrying CRISPR-resistant synonymous mutations was generated and cloned into the pLHCX using the HiFi assembly mix (NEB) according to the manufacturer's guidelines. VLPs were gener-ated by transfecting HEK293T cells with pLHCX plasmids, along with plasmids encoding VSV-G and the murine leukemia virus Gag-Pol. At 48h after transfection, supernatants were collected, clarified by centrifugation and filtered through a 0.22µm filter. HeLa ZAP$^{-/-}$ or A549 ZAP$^{-/-}$ cells were seeded onto 12-well plates and were transduced with 500µL of super-natants in 5µg/mL of polybrene. Cells were moved to selection medium containing 500µg/mL hygromycin. Expression of transduced genes was validated by western blotting. To generate the STAT1 reporter cell line, HEK293T were transfected with pLminP_Luc2P_RE57 [78], and plasmids encoding HIV-1 Gag-pol and VSV-G and, 48h later, supernatants were used to transduce A549 cells. Single cell clones from transduced cells were generated by limiting dilution and selected for the expression of EGFP.

### Western blotting

Cells were lysed in NuPAGE LDS sample buffer (Invitrogen) containing 1% β-mercaptoethanol, clarified by syringe-passaging and reduced in 72˚C for 20min. Cell lysates were then resolved onto a 4–12% polyacrylamide NuPAGE 4–12% Bis-Tris gels (Invitrogen) in MOPS running buffer (50mM MOPS, 50mM Tris, 3.5 mM SDS and 1 mM EDTA). Proteins were then transferred to nitrocellulose membranes using Tris-Glycine transfer buffer (25mM Tris, 192 mM

Glycine), blocked in BlockOut blocking buffer (Rockland) and incubated with the following antibodies: anti-tubulin (T5168 Sigma, 1:10,000, overnight, 4˚C), anti-HA (600-401-384 Rockland, 1:10,000, overnight, 4˚C), anti-HA (16b12 BioLegend, 1:10,000, 2h, room temperature), anti-FLAG (F3165 Sigma, 1:5,000, 2h, room temperature), anti-TRIM25 (12573–1-AP ProteinTech, 1:5,000, overnight, 4˚C), anti-ZAP (16820–1-AP ProteinTech, 1:10,000, overnight, 4˚C). Membranes were then washed in PBS containing 0.2% Tween-20 and blotted with fluorescent anti-rabbit (Licor) or anti-mouse (Licor) antibodies for 45min at room temperature. Membranes were imaged using a Licor Odyssey and analyzed using EmpiriaStudio software.

### Co-immunoprecipitation

HEK293T or HEK293T ZAP$^{-/-}$ TRIM25$^{-/-}$ cells transfected with indicated plasmids and incubated at 37˚C for 48h using lipofectamine 2000 according to the manufacturer's guidelines (Invitrogen). Supernatants were then removed and cells were then lysed in 1.5 mL of ice-cold lysis buffer (10mM HEPES pH7.5, 30mM KCl, 40µM EDTA, 0.1% Igepal CA-630, 1mM DTT supplemented with a cocktail of protease inhibitors) on ice for 15min. Cell lysates were clarified by centrifugation at 14,000xg for 10min at 4˚C, before incubating with Protein G Dynabeads pre-absorbed with anti-FLAG (F3165 Sigma) or anti-HA (600-401-384 Rockland) antibodies) for 2h at 4˚C on a rotating wheel. Beads were then washed twice in Lysis Wash buffer (10mM HEPES pH7.5, 30mM KCl, 40µM EDTA, 0.1% Igepal CA-630, 1mM DTT) followed by two washes in IP wash buffer (50mM HEPES pH 7.5, 300mM KCl, 2mM EDTA, 0.5% Igepal CA-630, 1mM DTT). Beads were then resuspended in 50µL of LDS NuPAGE loading buffer containing 1% β-mercaptoethanol and reduced at 72˚C for 20min. Protein complexes were then analyzed by SDS-PAGE/western blotting.

### Transfection with short-interfering RNAs (siRNAs)

Pools of four siRNA per gene targeting ZAP (ZC3HAV1, L-017449-01-0005), TRIM25 (L-006585-00-0005), Riplet (RNF135, L-007087-00-0005), RIG-I (DDX58 L-012511-00-0005) or non-targeting controls (D-001810-10-05) were obtained from Horizon Discovery and diluted in water to 10µM solutions. For each transfection, 5µL of a 10µM siRNAs stock solution was resuspend in Opti-MEM and incubated with Lipofectamine RNAiMAX transfection reagent (ThermoFisher) according to the manufacturer's guidelines. The transfection mix was then added to a well of 6-well plate and incubated for 10min at room temperature. For each well, 3e5 cells were reverse transfected for 48h before being used in experiments. The efficiency of siRNA knockdown was assessed by western blotting, when antibodies were available, or quantitative PCR for Riplet.

### Stimulation with immune sensor agonists

To simulate RIG-I, 5µM of 5' triphosphate hairpin RNA (3p-hRNA, Invivogen) was incubated with lipofectamine for 10min in Opti-MEM and then transferred to cells. Cells were then incubated for 8h at 37˚C, before lysed and total RNA was extracted as previously described. Similarly, 1µg/mL of poly I:C (P1530 Sigma) was resuspended in Opti-MEM in the presence of lipofectamine and then transferred to cells for 8h at 37˚C. Cells were lysed in RA1 buffer and RNA extracted as before. Extracted RNA reverse transcribed using the Superscript IV First Strand Synthesis kit (Invitrogen), using random hexamers, following the protocol supplied by the manufacturer. *IFNB1*, *IFNL2* and *GAPDH* transcripts were quantified by quantitative RT-PCR using the TaqMan gene expression assay (Hs01077958_s1 Human IFNB1, Hs02786624_g1 Human GAPDH, Hs00820125_g1 Human IFNL2, ThermoFisher) according to manufacturer's protocol.

### Protein crosslinking

Cells were seeded onto 12-well plates and incubated for 24h at 37˚C. Supernatants were then replaced by 1mL of serum-free DMEM containing 0.00625% glutaraldehyde (Tokyo Chemical Industry) for 15min at room temperature. Supernatants

were then removed, remaining glutaraldehyde was quenched in 50mM Tris-HCL pH 8 (in PBS) for 10min at room temperature and lysed in LDS NuPAGE buffer. Cell lysates were clarified and reduced as before and then resolved onto NuPAGE Tris-Acetate 3–8% gels in MOPS running buffer. Protein samples were then transferred to nitrocellulose membranes at 13V for 16-18h. Membranes were prepared for western blotting as described before.

### In vitro RNA transcription

Self-amplifying RNA (saRNA) encoding firefly luciferase was produced using in vitro transcription from plasmids encoding an saRNA replicon that contains the alphaviral non-structural genes (NSP1 - NSP4) - isolated from the Venezuelan equine encephalitis virus (VEEV), including the 5'UTR, sub-genomic promoter and 3'UTR - and the firefly luciferase gene luciferin-4-monooxygenase, derived from the common eastern firefly (*Photinus pyralis,* UniProt ID: P08659). The pDNA was transformed into *E. coli* (New England BioLabs, UK), cultured in 100 mL of Luria Broth (LB) with 100 µg mL−1 Kanamycin sulphate (Sigma Aldrich, UK). Plasmid was purified using a Plasmid Plus MaxiPrep kit (QIAGEN, UK) and the concentration and purity was measured on a NanoDrop One (ThermoFisher, UK). pDNA was linearized using SapI for 2 h at 37 °C, followed by a second linearisation also with fresh SapI for 1 h 37 °C. Capped in vitro RNA transcripts were produced using 1 µg of linearized DNA template in a HiScribe reaction (New England BioLabs, UK) for 2 h at 37 °C, according to the manufacturer's protocol. The RNA cap (CleanCap AU, TebuBio, UK) was included within the HiScribe IVT reaction. Transcripts were then purified by overnight LiCl precipitation at −20 °C, centrifuged at 14,000 RPM for 20 min at 4 °C to pellet, washed with 70% EtOH, centrifuged at 14,000 RPM for 10 min at 4 °C and resuspended in UltraPure H2O (Ambion, UK) and stored at −80 °C until further use. SaRNA was produced from two separate plasmid templates, the first that coded for the WT VEEV and Luciferase which has high numbers of CpG dinucleotides, and the second coding for VEEV plus Luciferase that had 90% of CpG motifs removed.

### Lipid nanoparticle formulation of self-amplifying RNA

Lipid nanoparticle (LNP_ formulations were prepared as previously described in [79]. Briefly, C12-200, 1,2-distearoyl-sn-glycero-3-phosphocholine (DSPC), cholesterol (plant-derived) and 1,2-dimyristoyl-sn-glycero-3-phosphoethanolamine-N-[methoxy(polyethylene glycol)-2000] (ammonium salt) (DMPE-PEG2000, Avanti Polar Lipids), and stock solutions were prepared in absolute ethanol. Lipids were mixed at molar ratios of 35:16:46.5:2.5 C12-200:DSPC:Cholesterol:DMPE-PEG2000 at a total lipid concentration of 15 mM. SaRNA solutions were prepared in 50 mM sodium acetate (Sigma-Aldrich) and 100 mM sodium chloride buffer (Sigma-Aldrich), adjusted to pH 5.5. LNPs were formulated using the NanoAssemblr Ignite nanoparticle formulation system (Cytiva) at an RNA-to-lipid ratio of 3:1 v/v (1:55 w/w) and a total flow rate of 8 mL/min. Formulated LNPs were diluted in Dulbecco's phosphate-buffered saline (DPBS, Gibco) at a ratio of 1:5 v/v and subsequently concentrated using Amicon Ultra-15 centrifugal filters with a 10 kDa molecular weight cutoff (Merck). LNPs were stored at 4 °C for immediate use or at −80 °C in 20 mM Tris-HCl buffer (pH 7.5) with 10% sucrose for long-term storage. The concentration of encapsulated RNA was quantified using the Quant-iT RiboGreen RNA Assay Kit (Thermo Fisher Scientific) according to the manufacturer's instructions. Fluorescence intensity was measured with a FLUOstar Omega plate reader (BMG Labtech) at excitation/emission wavelengths of 485/535 nm. LNP size, polydispersity index (PDI), and zeta potential were determined using a Zetasizer Nano ZS (Malvern Instruments).

### Delivery of saRNA

For luciferase activity experiments, A549 cells were seeded onto a 96-well plate and, on following day, cells were transfected with 50ng of encapsulated saRNA in LNP or 50ng of saRNA complexed with lipofectamine in Opti-MEM according to the manufacturer's guidelines. After 24h, supernatants were removed, and cells were lysed in 50µL of passive lysis buffer. Firefly luciferase activity was measured using the Bright-Glo luciferase assay system (Promega). To assess IFN

production from saRNA delivery, A549 cells were seeded onto a 24-well plate and transfected on the following day with 150ng of saRNA, as before. At 8h post-transfection, cells were lysed in RA1 solution and total RNA was extracted using the Nucleospin RNA kit. RNA samples were processed and *IFNB1* transcripts were quantified as described above. To assess STAT1 activity, A549-STAT1 reporter cells were seeded onto a 96-well plate and 24h later, 50ng of saRNA encoding EGFP was delivered either via LNP or lipofectamine. STAT1 reporter activity was measured 24h later by measuring firefly luciferase activity using the BrightGlo Luciferase Assay System (Promega).

### Protein structure predictions

Protein sequences from all known human DNA/RNA sensors and downstream effectors (Fig 1b, "bait" proteins) as well as all human SPRY-domain containing proteins annotated in InterPro (Fig 1b, "prey" proteins) were retrieved from Uniprot and used to generate one-to-one structure predictions in AlphaFold3 [80]. The predicted aligned error (PAE) maps of the highest scored protein structure prediction were then used to calculate an interaction score. For each protein pair, the local PAE values between residues of bait and prey proteins were reversed and and the grand sum of this matrix was called observed contacts (OC). To evaluate how observed interaction scores vary with protein length, we perform a proteome-wide interaction screen between using ZAP as a bait. From this screen, we observed that larger proteins tend to yield larger observed contacts. Based on this preliminary proteome-wide screen, we calculated expected contacts (EC) for each protein using the following formula: $EC = 243.9 \times length + 279,484$. Interaction scores were then calculated by the ratio of EC over OC. Interaction scores for each protein pair can be found in S1 Data. Scripts used to calculate interaction scores and to plot PAE maps can be found on github.com/DanSallves/ProteinInteraction/

### Phylogenetic and synteny analyses

The protein sequences of human TRIM25 and Riplet/RNF135 were retrieved from Uniprot and used to identify orthologues in other species using the NCBI's Blastp suite. Retrieved sequences with significant E-values were then used in phylogenetic analyses. Protein sequences from representative vertebrate taxa were aligned using Clustal Omega under default settings. The resulting multiple sequence alignment was used directly for phylogenetic inference in IQ-TREE v3.0.1 [81]. The best-fitting substitution model (JTTDCMUT+I+R4) was selected by ModelFinder according to the Bayesian Information Criterion (BIC). A maximum-likelihood phylogeny was reconstructed, and branch support was evaluated with 1,000 ultrafast bootstrap replicates and the SH-aLRT test with 1,000 replicates. The final tree was visualized and annotated with iTOL v7. For synteny studies, *TRIM25* and *RNF135/Riplet* loci from indicated species were analyzed using the Ensembl, Genomicus Browser and the NCBI Genome Viewer.

### Supporting information

**S1 Data. Contains the interaction scores for all SPRY-domain containing proteins and indicated sensors and/or effectors.**
(ZIP)

**S1 Fig. Analysis of all SPRY-Sensor predictions generated by AlphaFold.** Proportion of known biological functions of all human SPRY-containing proteins (a). Distribution of interaction scores across all SPRY-sensors pairs (b). SPRY-containing proteins with the largest number of interactor partners in the AlphaFold prediction screen (c). PAE maps of selected sensors and SPRY-domain containing proteins (d). HEK293T cells were transfected with plasmids encoding OAS1, wildtype TRIM58 or a catalytically inactive mutant of TRIM58 (TRIM58 C57S C60S). Cells were lysed 48h after transfected and OAS1 levels was measured by western blotting analysis (e). Multiple sequence alignment of the RING domains of Riplet, TRIM25, TRIM21 and TRIM58 (f).
(TIFF)

**S2 Fig. Comparison of full-length and SPRY-only AlphaFold prediction.** Frequency distribution of Spearman's correlation coefficients between Full-Length and SPRY-only AlphaFold prediction datasets (a). Comparison of interaction scores obtained from Full-Length and SPRY-only datasets for cGAS, RIG-I, OAS1 and ZAP (b). Schematic represasion of the topological organization of hnRNPUL2: an N-terminal DNA-binding SAP domain, a central SPRY domain and a C-terminal AAA/NTPase-like domain (c). PAE maps of hnRNPUL2 full-length and SPRY only interactions with cGAS (d). Comparison of interaction scores obtained from Full-Length and SPRY-only datasets for TRIM16 and TRIM16 (e). PAE maps of ASH2L:MDA5 and TRIM51:MDA5 predicted interactions (f).
(TIFF)

**S3 Fig. Synteny of *TRIM25* and *Riplet*/*RNF135* across vertebrates.** Diagram of the genetic organization of TRIM25 and Riplet loci (in yellow) in vertebrate genomes (a). Mb, megabase (1 million base pairs). MYA, Million years ago.
(TIFF)

**S4 Fig. TRIM25 and Riplet support ZAP and RIG-I activity, respectively.** TRIM25 and Riplet were depleted from A549 cells using siRNAs and 48h later cells were analyzed by western blotting (a) and qPCR (b). Riplet and TRIM25 were depleted from A549 cells using siRNAs and 48h later cells were transfected with 3p-hRNA or poly I:C. After 8h after transfections, RNA was extracted and IFNL2 was quantified by qPCR (c). Riplet and TRIM25 were knocked-down from HeLa cells using siRNAs; cells were then transfected with 3p-hRNA or poly I:C and, after 8h, RNA was extracted and *IFNB* and *IFNL2* transcripts were quantified by qPCR (d). *IFNB1* and *IFNL2* fold-induction in HeLa and A549 cells treated with 3phRNA at 8h post-transfection (e). ZAP was depleted from HeLa cells (f) or A549 cells (g) using siRNAs. After 48h, cells were analyzed by western blotting or infected with a ZAP-sensitive EV71. Luciferase activity was measured 24h later. Riplet- or TRIM25-depleted HeLa cells were infected with ZAP-sensitive nLuc-EV71, and 24h later luciferase activity was quantified (h). Sequencing trace of the RNF135 exon 1 genomic locus in wildtype of Riplet$^{-/-}$ A549 cells (i). A549 Riplet$^{-/-}$ cells were transduced with retroviral vectors encoding the indicated HA-tagged proteins. Protein expression was measured by western blotting (j).
(TIFF)

**S5 Fig. TRIM25 expression impacts ZAP protein levels.** Structural representation of AlphaFold prediction of Riplet, TRIM25 full-length protein and the Riplet-T25$^{SPRY}$ and TRIM25-Riplet$^{SPRY}$ chimeric proteins (a) HEK293T TRIM25$^{-/-}$ ZAP$^{-/-}$ were transfected with increasing amounts of plasmids encoding TRIM25-HA or Riplet-HA and ZAP-L expression was analyzed by western blotting (b). A549 Riplet$^{-/-}$ cells were transduced with retroviral vectors encoding the indicated HA-tagged proteins. Protein expression was measured by western blotting (c).
(TIFF)

**S6 Fig. Structure of the SPRY domain of TRIM25 with variable loops highlighted in purple (a).** Multiple sequence alignment of orthologues of TRIM25, Riplet, ZAP and PARP12 (b). Predicted structure of ZAP-TRIM25 showing electrostatic surface (c). Inset showed positively charged R544 residue of TRIM25 accommodated by a negatively charged pocket formed by residues D3 and E5 of ZAP (c). HEK293T ZAP-/- TRIM25-/- cells were transfected with plasmid encoding wildtype or qMut ZAP-S along with TRIM25 followed by co-immunoprecipitation of ZAP-S and western blotting (d). Infection of ZAP$^{-/-}$ cells reconstituted with wildtype of qMut ZAP-S (e). Expression levels of single mutants of TRIM25 as indicated (f). HEK293T ZAP$^{-/-}$ TRIM25$^{-/-}$ cells were transfected with plasmid encoding the indicated mutants of ZAP-S and TRIM25 followed by co-immunoprecipitation of ZAP-S and western blotting (g). HEK293T ZAP$^{-/-}$ TRIM25$^{-/-}$ cells were transfected with plasmid encoding wildtype or qMut ZAP-S along with KHNYN followed by co-immunoprecipitation of ZAP-S and western blotting (h). HEK293T were transfected with plasmids encoding ZAP-L-HA, GFP-HA and OASL; cell lysates were used to purified ZAP- or GFP-complexes using an anti-HA antibody and analyzed by western blotting (i). HEK293T ZAP$^{-/-}$ TRIM25$^{-/-}$ cells were transfected with plasmid encoding wildtype or qMut ZAP-S along with OASL followed

by co-immunoprecipitation of ZAP-S and western blotting (j). Structure of the RNA-binding domain of ZAP (in blue) and the C-terminal domain of KHNYN (red, KHNYN CTD, obtained from PDB:9BGL) and the predicted stucture of OASL (in green) (k). Predicted structured of the RNA-binding domain of ZAP (light blue) with residues known to interact with KHNYN in red (described by Bohn et al. [25]) and residues predicted to interact with TRIM25 in blue (l). (TIFF)

## Acknowledgments

We would like to acknowledge advice and feedback from members of the Group of Mucosal Infection and Immunity at Imperial College London. The authors would also like to acknowledge Professor John Tregoning and Dr Zying Wang (Imperial College London) for donating total RNA extracted from lung samples from BALB/c mice, Professor Kevin James (University of Viriginia) for gifting the pLX302 OAS1-V5 puro plasmid, Professor Nan Yan (University of Texas Southwestern Medical Center) for sharing the pMRX-hSTING-EGFP plasmid and Professor Ramnik Xavier (Broad Institute) for donating the pLminP_Luc2P_RE57 plasmid. The authors would like to thank Richard Liebowitz, Jane Reader, Eldrian Tho and Sijung Lee for help with AlphaFold predictions.

## Author contributions

**Formal analysis:** Ibrahim Syed, Sheng Chen, David J Peeler, Paul F McKay, Marco A Briones-Orta, Jennifer A Bohn, Robin J. Shattock, Daniel Gonçalves-Carneiro.

**Funding acquisition:** Robin J. Shattock, Daniel Gonçalves-Carneiro.

**Investigation:** Ibrahim Syed, Sheng Chen, David J Peeler, Paul F McKay, Marco A Briones-Orta, Daniel Gonçalves-Carneiro.

**Supervision:** Robin J. Shattock, Daniel Gonçalves-Carneiro.

**Writing – original draft:** Daniel Gonçalves-Carneiro.

**Writing – review & editing:** Ibrahim Syed, Sheng Chen, David J Peeler, Paul F McKay, Marco A Briones-Orta, Jennifer A Bohn, Daniel Gonçalves-Carneiro.

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
