## [Editor Report · Decision Letter 0]

27 Jan 2026

PPATHOGENS-D-26-00176

Specialization of ubiquitin ligases to distinct nucleic acid sensors

PLOS Pathogens

Dear Dr. Goncalves-Carneiro,

Thank you for submitting your manuscript to PLOS Pathogens. After careful consideration, we feel that it has merit but does not fully meet PLOS Pathogens's publication criteria as it currently stands. Therefore, we invite you to submit a revised version of the manuscript that addresses the points raised during the review process.

We look forward to receiving your revised manuscript.

Kind regards,

Benjamin G Hale, PhD

Guest Editor

PLOS Pathogens

Michael Letko

Section Editor

PLOS Pathogens

Sumita Bhaduri-McIntosh

Editor-in-Chief

PLOS Pathogens

orcid.org/0000-0003-2946-9497

Michael Malim

Editor-in-Chief

PLOS Pathogens

orcid.org/0000-0002-7699-2064

**Additional Editor Comments:**

Thank-you for providing a clear revision plan to address the positive and constructive expert reviews from Reviews Commons. Please proceed with modifying the manuscript as you describe with both text edits and the results from the new experiments you propose.

**Journal Requirements:**

https://journals.plos.org/plospathogens/s/submission-guidelines#loc-parts-of-a-submission

- ® on page: 16

- TM on page: 16.

5) We have noticed that you have uploaded Supporting Information files, but you have not included a list of legends. Please add a full list of legends for your Supporting Information files after the references list.

6) When completing the data availability statement of the submission form, you indicated that you will make your data available on acceptance. We strongly recommend all authors decide on a data sharing plan before acceptance, as the process can be lengthy and hold up publication timelines. Please note that, though access restrictions are acceptable now, your entire data will need to be made freely accessible if your manuscript is accepted for publication. This policy applies to all data except where public deposition would breach compliance with the protocol approved by your research ethics board. If you are unable to adhere to our open data policy, please kindly revise your statement to explain your reasoning and we will seek the editor's input on an exemption. Please be assured that, once you have provided your new statement, the assessment of your exemption will not hold up the peer review process.

7) Please amend your detailed Financial Disclosure statement. This is published with the article. It must therefore be completed in full sentences and contain the exact wording you wish to be published.

8) Please send a completed 'Competing Interests' statement, including any COIs declared by your co-authors. If you have no competing interests to declare, please state "The authors have declared that no competing interests exist". Otherwise please declare all competing interests beginning with the statement "I have read the journal's policy and the authors of this manuscript have the following competing interests"

**Reviewers' Comments:**

**Figure resubmission:**
---

## [Decision Letter · Decision Letter 1]

23 Apr 2026

Dear Dr Goncalves-Carneiro,

We are pleased to inform you that your manuscript 'SPRY domains encode ubiquitin ligase specificity for ZAP and RIG-I' has been provisionally accepted for publication in PLOS Pathogens.

Before your manuscript can be formally accepted you will need to complete some formatting changes, which you will receive in a follow up email. A member of our team will be in touch with a set of requests. Please also take note of the request by reviewer 1 that you should update the figures to add molecular weight markers to all western blots, either at this stage or at latest during proofing.

Best regards,

Benjamin G Hale, PhD

Academic Editor

PLOS Pathogens

Michael Letko

Section Editor

PLOS Pathogens

Sumita Bhaduri-McIntosh

Editor-in-Chief

PLOS Pathogens

orcid.org/0000-0003-2946-9497

Michael Malim

Editor-in-Chief

PLOS Pathogens

orcid.org/0000-0002-7699-2064

The AE and one of the original reviewers have both reviewed the revised manuscript, and are satisfied with how the original reviewer comments have been addressed with new data and clarifications. Prior to final publication, and at latest during the proofing stage, please update the figures to add molecular weight markers to all western blots (as raised by reviewer 1).

Reviewer Comments (if any, and for reference):

Reviewer's Responses to Questions

**Part I - Summary**

Reviewer #1: The authors have addressed my comments, concerns and questions with new data and clarifications to the text and figures. This is a very interesting and well conducted study, executed to a high standard, which raises a number of interesting hypotheses for further investigation.

**Part II – Major Issues: Key Experiments Required for Acceptance**

Reviewer #1: No further experiments requested.

**Part III – Minor Issues: Editorial and Data Presentation Modifications**

Reviewer #1: Molecular weights need to be added to all Western blots.

PLOS authors have the option to publish the peer review history of their article (what does this mean?). If published, this will include your full peer review and any attached files.

Reviewer #1: **Yes:** Adam Fletcher

---

## [Editor Report · Acceptance letter]

Dear Dr Gonçalves-Carneiro,

We are delighted to inform you that your manuscript, "SPRY domains encode ubiquitin ligase specificity for ZAP and RIG-I," has been formally accepted for publication in PLOS Pathogens.

Best regards,

Sumita Bhaduri-McIntosh

Editor-in-Chief

PLOS Pathogens

orcid.org/0000-0003-2946-9497

Michael Malim

Editor-in-Chief

PLOS Pathogens

orcid.org/0000-0002-7699-2064